# WaveBench: Benchmarking Data-driven Solvers for Linear Wave Propagation PDEs

**Tianlin Liu**\* *University of Basel*

**Jose Antonio Lara Benitez**\* *Rice University*

**Florian Faucher**\* *Team Makutu, Inria, University of Pau and Pays de l'Adour, TotalEnergies, CNRS, France*

**AmirEhsan Khorashadizadeh** *University of Basel*

**Maarten V. de Hoop** *Rice University*

**Ivan Dokmanić** *University of Basel*

\**These authors contributed equally to the work.*

**Reviewed on OpenReview:** *https://openreview.net/forum?id=6wpInwnzs8*

## Abstract

Wave-based imaging techniques play a critical role in diverse scientific, medical, and industrial endeavors, from discovering hidden structures beneath the Earth's surface to ultrasound diagnostics. They rely on accurate solutions to the forward and inverse problems for partial differential equations (PDEs) that govern wave propagation. Surrogate PDE solvers based on machine learning emerged as an effective approach to computing the solutions more efficiently than via classical numerical schemes. However, existing datasets for PDE surrogates offer only limited coverage of the wave propagation phenomenon. In this paper, we present WaveBench, a comprehensive collection of benchmark datasets for wave propagation PDEs. WaveBench **(1)** contains 24 datasets that cover a wide range of forward and inverse problems for time-harmonic and time-varying wave phenomena in 2D; **(2)** includes a user-friendly PyTorch environment for comparing learning-based methods; and **(3)** comprises reference performance and model checkpoints of popular PDE surrogates such as U-Nets and Fourier neural operators. Our evaluation on WaveBench demonstrates the impressive performance of PDE surrogates on *in-distribution* samples, while simultaneously unveiling their limitations on *out-of-distribution* (OOD) samples. This OOD-generalization limitation is noteworthy, especially since we use stylized wavespeeds and provide abundant training data to PDE surrogates. We anticipate that WaveBench will stimulate the development of accurate wave-based imaging techniques through machine learning.

## 1 Introduction

Waves are behind imaging modalities as diverse as reflection seismology, medical ultrasound, and X-ray crystallography. Imaging with waves relies on mathematical models of wave propagation expressed through partial differential equations (PDEs) called wave equations.

Since conventional numerical PDE solvers are computationally expensive for large-scale problems, recent research has witnessed a rapid emergence of machine learning-based models to approximate PDE solutions (Li et al., 2020; Nelsen & Stuart, 2021; Lu et al., 2019; Bhattacharya et al., 2021; Li et al., 2021; Huang et al., 2021; Wang et al., 2021; Gupta et al., 2021; de Hoop et al., 2022a; Kissas et al., 2022; Brandstetter et al.,

2023). These models, collectively referred to as *PDE surrogates*, are trained using a dataset of ground-truth PDE solutions. Properly trained PDE surrogates can offer faster inference speeds compared to traditional numerical solvers while maintaining accuracy. Realizing this advantage, however, typically requires having access to high-quality PDE solutions for training purposes.

Indeed, it is hard to overstate the significance of high-quality datasets in the success of modern machine learning models in computer vision and natural language processing. Similarly, in the case of PDE surrogates, the importance of high-quality datasets has become evident. Recent efforts focus on constructing high-quality datasets for PDE surrogates (Lu et al., 2022; de Hoop et al., 2022b; Gupta & Brandstetter, 2022; Takamoto et al., 2022). These datasets cover a wide range of equations, especially those related to fluid dynamics, such as Darcy flow, shallow-water, and Navier–Stokes equations (Li et al., 2021; Gupta & Brandstetter, 2022; Stachenfeld et al., 2022; Holl et al., 2020). However, a comprehensive dataset for a broad family of wave PDEs is still missing.

To address the gap, we present WAVEBENCH, an extensive collection of benchmark datasets designed for wave propagation PDEs. WAVEBENCH includes 24 datasets, encompassing two categories of forward and inverse problems of acoustic waves: *time-harmonic problems* and *time-varying problems*. These datasets are constructed using open-source software tools `hawen` (Faucher, 2021) for time-harmonic waves and `j-wave` (Stanziola et al., 2023) for time-varying waves. We have made these datasets publicly accessible for researchers to access. Moreover, we provide a PyTorch (Paszke et al., 2019) environment that enables easy training and comparison between various PDE surrogate models. We also include checkpoints and reference results for popular PDE surrogates of Fourier neural operators (Li et al., 2021), U-Nets (Ronneberger et al., 2015), and the U-shaped neural operator (Rahman et al., 2023). By providing these resources, we aim to foster the development of machine-learning techniques for wave imaging.

## 2 Background and related work

**ML methods to approximate operators described by PDEs.** The solution map of a generic PDE can be written as an operator $G^\dagger : \mathcal{A} \to \mathcal{U}$, where $\mathcal{A}$ and $\mathcal{U}$ are normed function spaces defined on some bounded subsets in $\mathbb{R}^d$. We focus on 2D domains, thus $d = 2$. For instance, $G^\dagger$ can be a mapping that converts PDE coefficients $a \in \mathcal{A}$ into a PDE solution $G^\dagger(a) \in \mathcal{U}$. A PDE surrogate $G_\theta$ is a data-driven emulator of the true PDE solution map $G^\dagger$, with $\theta$ denoting the model parameters. A PDE surrogate is learned from a training dataset of input–output pairs $\{(a_j, u_j)\}_{j=1}^N$, where $a_j \in \mathcal{A}$ and $u_j = G^\dagger(a_j)$ are prepared using conventional numerical solvers. The *loss* quantifying how well a PDE surrogate with parameters $\theta$ fits the data is formulated as

$$L_N(\theta) := \frac{1}{N} \sum_{j=1}^{N} l\Big(G_\theta(a_j), u_j\Big),$$

where the error function $l(\cdot, \cdot)$ is often chosen as the absolute error $l(\widehat{u}, u) = \|\widehat{u} - u\|_{\mathcal{U}}$ or the relative error $l(\widehat{u}, u) = \|\widehat{u} - u\|_{\mathcal{U}} \|u\|_{\mathcal{U}}^{-1}$ (Kovachki et al., 2023). Once trained, the model $G_\theta$ yields approximate PDE solutions, often faster than traditional solvers.

**PDEs surrogates.** A PDE surrogate $G_\theta$ can be parameterized in various ways. Popular choices include kernel-based models (Kadri et al., 2016; Griebel & Rieger, 2017), random feature models (Nelsen & Stuart, 2021), Gaussian processes Chen et al. (2021); Harkonen et al. (2022); Henderson et al. (2023), and neural networks (Lu et al., 2019; Kovachki et al., 2023; Bhattacharya et al., 2021; Wang et al., 2021; Kissas et al., 2022; Brandstetter et al., 2023). Among them, neural networks represent the current state-of-the-art in empirical performance. Recently, Lu et al. (2022); de Hoop et al. (2022b); Gupta & Brandstetter (2022); Takamoto et al. (2022) ran a comprehensive numerical comparison of various models, showing their relative merits in different scenarios. For PDEs discretized on 2D grid meshes, Fourier neural operators (FNO) and U-Nets are highly performant (Takamoto et al., 2022; Gupta & Brandstetter, 2022); we thus adopt them as baselines.

**Benchmarks and datasets for PDEs surrogates.** Recent advancements in PDE surrogates have led to the development of standardized datasets that serve as benchmarks (Lu et al., 2019; Kothari et al., 2020;

Li et al., 2021; Lu et al., 2022; de Hoop et al., 2022b; Gupta & Brandstetter, 2022; Takamoto et al., 2022; Benitez et al., 2023). These datasets comprise a wide range of PDEs, usually with particular emphasis on fluid dynamics problems such as Darcy flow, shallow-water, and Navier–Stokes problems. Despite this impressive variety, there is a noticeable gap in the coverage of wave propagation PDEs. While a few work apply PDE surrogates on wave problems (Kothari et al., 2020; de Hoop et al., 2022b; Deng et al., 2022; Benitez et al., 2023), each focuses on a specialized setting. For instance, Kothari et al. (2020) and Deng et al. (2022) focus on certain time-varying wave problems, and Stanziola et al. (2021); de Hoop et al. (2022b) and Benitez et al. (2023) concentrate on time-harmonic ones. In contrast, the proposed WAVEBENCH provides a comprehensive range of wave propagation datasets by considering both time-varying and time-harmonic problems. Additionally, we cover a wide range of frequency configurations for time-harmonic problems and take different types of wave speeds into account for both time-varying and time-harmonic problems. To facilitate model comparisons, we also offer a unified PyTorch environment for comparing and benchmarking models.

## 3 WaveBench datasets

Our proposed WAVEBENCH consists of 24 datasets, divided into two problem categories: time-harmonic wave problems and time-varying problems. The time-harmonic wave problem is further decomposed into two with the consideration of acoustic and elastic waves.

### 3.1 Datasets for time-harmonic wave problems

#### 3.1.1 Time-harmonic acoustic waves.

We consider the propagation of time-harmonic acoustic waves (Martin, 2021; Colton et al., 1998; Faucher & Scherzer, 2020) written in terms of the pressure field $p$. Upon assuming a medium with constant density, it results into considering the Helmholtz equation (e.g., Faucher (2017)):

$$-\left(\Delta + \frac{\omega^2}{c(\boldsymbol{x})^2}\right) p(\boldsymbol{x}, \omega) = f(\boldsymbol{x}, \omega),\tag{1}$$

where $p = p(\boldsymbol{x}, \omega)$ is a pressure field at the angular frequency $\omega$, $c$ is a wavespeed function. Throughout this work, our domain is in 2D, that is, $\boldsymbol{x} \in \mathbb{R}^2$. For the source function $f$, we consider a delta-Dirac in space $\delta(\boldsymbol{y})$ where $\boldsymbol{y}$ is the position of the source. On the boundary of the domain, we follow a geophysical configuration (Benitez et al., 2023; Faucher & Scherzer, 2020): The boundary is divided into two distinct parts. The upper boundary corresponds to an interface $\Gamma_1$ where Dirichlet zero conditions (free surface) are enforced. In the remaining portion of the boundary, $\Gamma_2$ we assume *absorbing boundary conditions* to prevent waves from reflecting back to the medium, Engquist & Majda (1977). We have,

$$p(\boldsymbol{x}, \omega) = 0, \qquad \text{on } \Gamma_1 \text{ (free surface)},\tag{2a}$$

$$\left(\partial_\nu - \frac{i\omega}{c(\boldsymbol{x})}\right) p(\boldsymbol{x}, \omega) = 0, \qquad \text{on } \Gamma_2 \text{ (absorbing boundary conditions)}.\tag{2b}$$

**Experimental setup** We use the same domain size and a fixed point source (near surface) for all frequencies. To stay in the statistical learning setup, we randomly generate wavespeed as the composition of an affine transformation and a Gaussian random field with the Whittle–Matérn covariance as described in Benitez et al. (2023). Using the same notation as Benitez et al. (2023), we set the smoothness parameter $\nu$ of the field to be 1 for all the cases. Furthermore, we let the coefficients $\boldsymbol{\lambda} = (\lambda_x, \lambda_y)$ appear in Whittle–Matérn covariance (Benitez et al., 2023) be as follows: (a) for the isotropic case we choose $\boldsymbol{\lambda} = (0.1, 0.1)$, and (b) for the anisotropic case we have $\boldsymbol{\lambda} = (0.2, 0.5)$. We illustrate in Figure 1 a wavespeed $c$ and the corresponding pressure field $p$.

$$\text{Experiment config.} \begin{cases} \text{The domain is 2D having the size } 1.27{\times}1.27\text{km}^2 \\ 50\,000 \text{ GRF wave speeds generated, imposing } 1.5\text{km}\,\text{s}^{-1} \le c(x) \le 5\text{km}\,\text{s}^{-1} \\ \text{The data are } p \text{ that solve (1) at frequency } \omega/(2\pi) = 10,\,15,\,20 \text{ and } 40 \text{ Hz.} \end{cases}\tag{3}$$

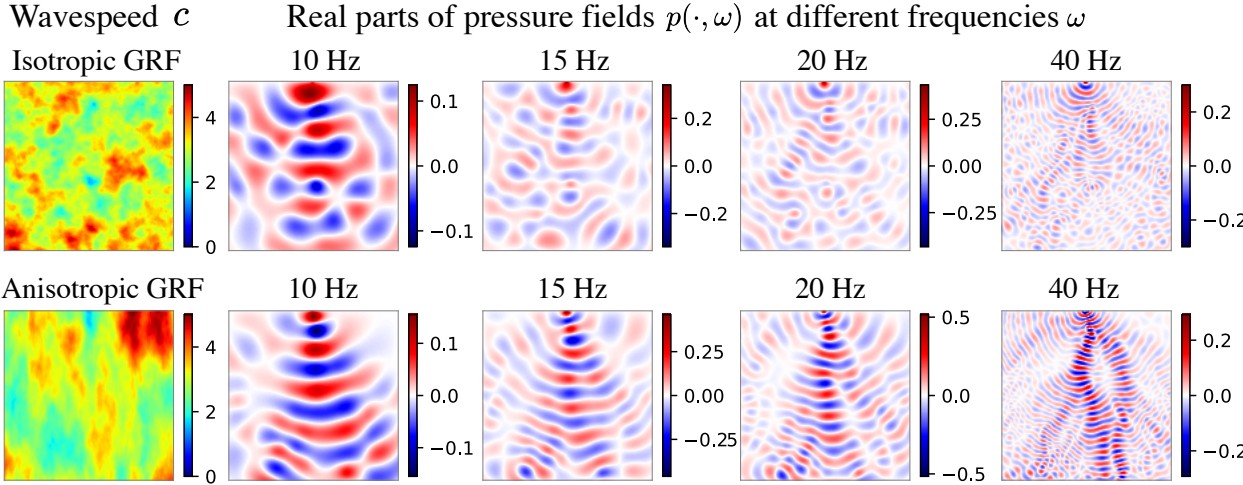

Figure 1: **Visualization of samples from the acoustic time-harmonic datasets**. At a fixed frequency $\omega$, the solver to the Helmholtz equation (1) converts a wavespeed $c$ to a pressure field $p(\cdot, \omega)$. Two wavespeed $c$ are shown in the left-most panels. They are realizations of Gaussian random field (GRF) with an isotropic kernel and an anisotropic kernel, respectively. The right panels display ground-truth pressure field $p(\cdot, \omega)$ at frequencies $\omega/(2\pi) = 10\text{Hz}, 15\text{Hz}, 20\text{Hz},$ and $40\text{Hz}$.

At a fixed frequency $\omega$ and source $f$, the parametric form of the Helmholtz equation describes an operator $G^\dagger_{\text{helm}}$ that maps the wavespeed $c$ to a pressure field $p$:

$$G^\dagger_{\text{helm}} : c \mapsto p(\cdot, \omega). \tag{4}$$

To estimate the operator $G^\dagger_{\text{helm}}$ from data, our dataset contains paired wavespeed and pressure field $\{(c_j, p_j)\}_j$. We let the wavespeed $c$ be realizations of Gaussian random fields (GRF) with an isotropic kernel or an anisotropic kernel. For each wavespeed $c$, the corresponding ground truth pressure field $p$ is obtained by solving the PDE using a hybridizable discontinuous Galerkin (HDG) method (Faucher & Scherzer, 2020) implemented in `hawen` package (Faucher, 2021). We produce 8 time-harmonic datasets corresponding to 2 types of GRF wavespeeds (isotropic and anisotropic), and 4 frequencies. Each dataset contains 49,000 training samples, 500 validation samples, and 500 test samples. For an overview of these time-harmonic datasets, see Table 1.

### 3.1.2 Time-harmonic elastic waves.

Unlike the acoustic wave problem which models a scalar pressure field, the elastic case works with the displacement *vector* field $\boldsymbol{u}$, written in two dimensions as $\boldsymbol{u} = [\boldsymbol{u}_x, \boldsymbol{u}_y]$. Following time-harmonic propagation, each component ($\boldsymbol{u}_x$ or $\boldsymbol{u}_y$) is complex-valued. Under elastic isotropy, the time-harmonic equation has the form, (Carcione, 2007; Faucher, 2017):

$$-\rho(\boldsymbol{x})\omega^2\boldsymbol{u}(\boldsymbol{x}) - \nabla\Big(\lambda(\boldsymbol{x})\nabla \cdot \boldsymbol{u}(\boldsymbol{x})\Big) - \nabla \cdot \Big(\mu(\boldsymbol{x})\Big[\nabla\boldsymbol{u}(\boldsymbol{x}) + \big(\nabla\boldsymbol{u}(\boldsymbol{x})\big)^\top\Big]\Big) = \boldsymbol{g}(\boldsymbol{x}), \tag{5}$$

where $\omega$ is the angular frequency and $\boldsymbol{g}$ the source. The isotropic elastic medium is characterized by its density $\rho(x)$, and the Lamé parameters $\lambda(x)$ and $\mu(x)$ (in particular, $\mu(x)$ is the shear modulus). In elastic media, two body waves propagate, the P-wave (compressional or primary wave) and S-wave (shear or secondary wave), Carcione (2007). Each wave is associated with a wavespeed, respectively $c_p(x)$ and $c_s(x)$, which can be used to characterize the medium as an alternative to the Lamé parameters, and are given by,

$$c_p(\boldsymbol{x}) := \sqrt{\frac{\lambda(\boldsymbol{x}) + 2\mu(\boldsymbol{x})}{\rho(\boldsymbol{x})}}, \qquad c_s(\boldsymbol{x}) := \sqrt{\frac{\mu(\boldsymbol{x})}{\rho(\boldsymbol{x})}}. \tag{6}$$

| | Wavespeed $c$ | Frequency $\omega/2\pi$ |
|---|---|---|
| Acoustic wave | Isotropic GRF | 10 Hz |
| | | 15 Hz |
| | | 20 Hz |
| | | 40 Hz |
| | Anisotropic GRF | 10 Hz |
| | | 15 Hz |
| | | 20 Hz |
| | | 40 Hz |
| Elastic wave | Anisotropic GRF | 10 Hz |
| | | 15 Hz |
| | | 20 Hz |
| | | 40 Hz |

Table 1: **Summary of 12 time-harmonic datasets.** Each dataset corresponds to a governing time-harmonic wave equation (acoustic or elastic), a type of wavespeed (isotropic GRF or anisotropic GRF) and frequency (10, 15, 20, or 40 Hz). **Each dataset consists of 49,000 training samples, 500 validation samples, and 500 test samples.** The roles of the wavespeed $c$ and frequency $\omega$ in time-harmonic wave propagation can be seen in the Helmholtz equation (1).

**Experimental setup**   Throughout our elastic wave experiments, we let P-wavespeeds $c_p$, the S-wavespeed $c_s$ be realizations of anisotropic GRFs while the density is kept to $\rho = 1$. More precisely, we first generate $c_p$ as a GRF with values between 2.5 and 5.5 km.s$^{-1}$, we then generate a GRF function $\mathfrak{c}$ with values between 0.35 and 0.50 that we use as a scaling to create $c_s = \mathfrak{c}\, c_p$. Therefore, $c_s$ is a randomly scaled version of $c_p$. This choice is motivated as $c_p$ and $c_s$ are physical properties of a medium that are expected to contain the same geometry of structures (e.g., $\mu$ and $\rho$ appear in both $c_p$ and $c_s$ in (6)). For boundary conditions, we follow the same configuration as for the acoustic case with absorbing boundary conditions on the lateral and bottom boundaries, and a free-surface condition for the upper surface where (representing the interface between the ground and the air), see, e.g., Faucher (2017). We visualize the wavespeeds and the real parts of the displacement fields at different frequencies in Figure 2.

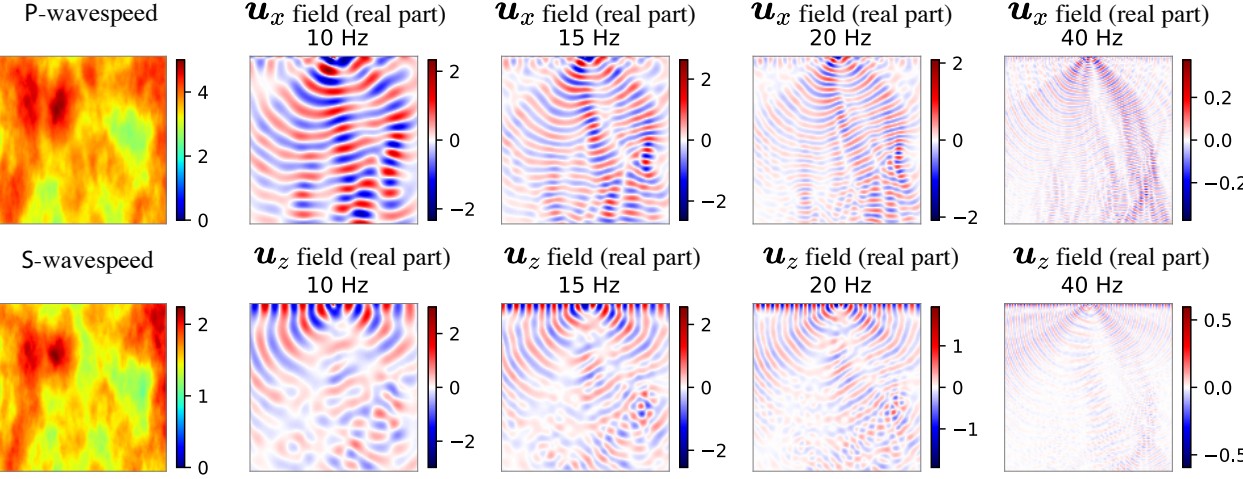

Figure 2: **Visualization of samples from the elastic time-harmonic datasets**. The wavespeeds are separated into P-waves and S-waves shown in the left-most column. At a fixed frequency $\omega$, the solver converts wavespeed $c$ to a displacement field $\boldsymbol{u} = [\boldsymbol{u}_x, \boldsymbol{u}_z]$. The right panels display these displacement fields at frequencies $\omega/(2\pi) = 10\text{Hz}, 15\text{Hz}, 20\text{Hz},$ and $40\text{Hz}$.

In the case of elastic propagation, for a fixed frequency $\omega$ and source $f$, the parametric form describes an operator $G^\dagger_{\text{elst}}$ that maps the P- and S-wavespeeds $c_p$ and $c_s$ to a displacement vector field $\boldsymbol{u}$:

$$G^\dagger_{\text{elst}} \, : \, (c_p, \, c_s) \, \mapsto \, \boldsymbol{u}(\cdot, \omega) := [\boldsymbol{u}_x(\cdot, \omega), \boldsymbol{u}_z(\cdot, \omega)]. \tag{7}$$

For the computation of $G^\dagger_{\text{elst}}$, similar to the acoustic case we use the open-source software `hawen` (Faucher, 2021). We list in Table 1 the frequencies used to generate the dataset.

## 3.2 Datasets for time-varying wave problems

We now turn to the governing equation for acoustic wave propagation in the time domain. The acoustic wave equation describes the evolution of pressure $q = q(\boldsymbol{x}, t)$ over time under the influence of the wavespeed $c = c(\boldsymbol{x})$:

$$\Delta q(\boldsymbol{x}, t) - \frac{1}{c(\boldsymbol{x})^2} \frac{\partial^2 q(\boldsymbol{x}, t)}{\partial t^2} = 0 \tag{8}$$

This equation is subject to radiating boundary conditions detailed in Appendix A. The time-varying quantity $q = q(\boldsymbol{x}, t)$ in (8) and the time-harmonic quantity $p = p(\boldsymbol{x}, \omega)$ in (1) are related: under appropriate assumptions, $q(\boldsymbol{x}, t)$ can be written as an integration of $p(\boldsymbol{x}, \omega)$ in the frequency domain. See Faucher (2017, Section 1.6) for a detailed derivation. We consider two problems that arise from the time-varying wave dynamics: the *reverse time continuation* and the *inverse source* problem.

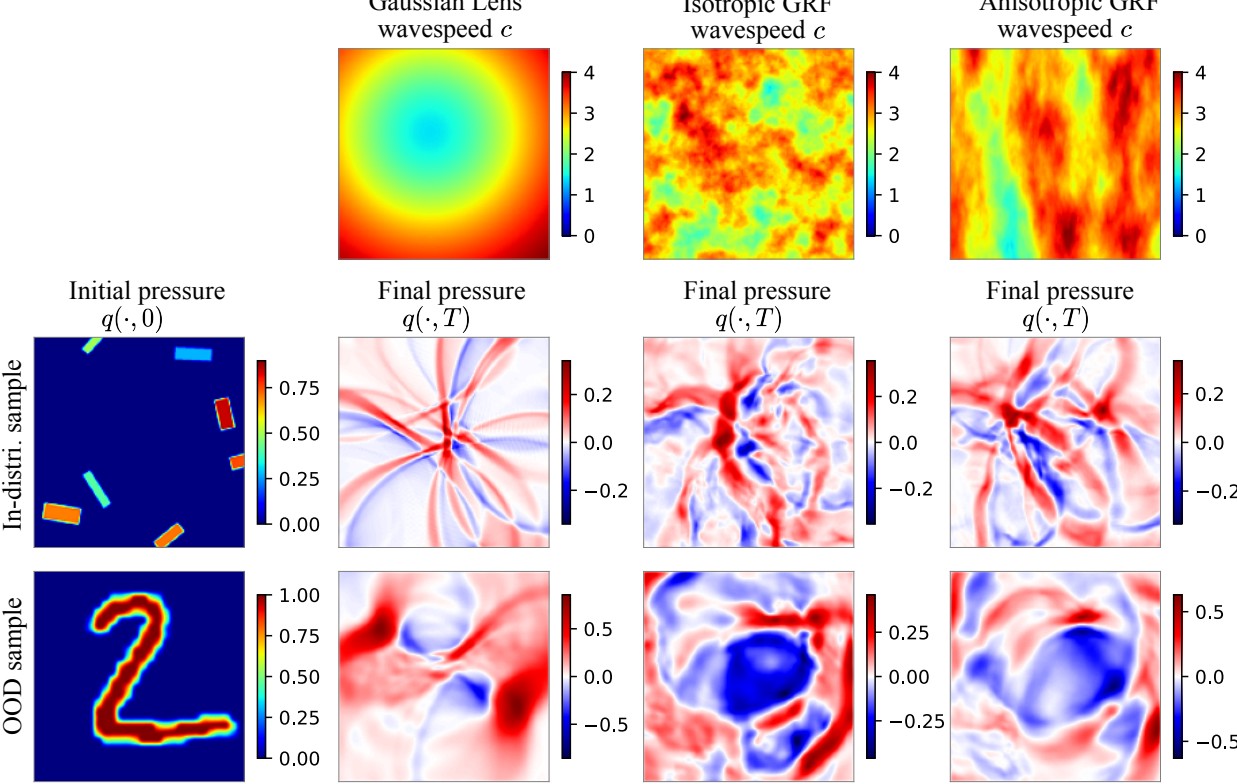

Figure 3: **Visualization of samples from the reverse time continuation (RTC) problems**. With a fixed wavespeed $c$, the goal of RTC is to map the final pressure $q(\cdot, T)$ to the initial pressure $q(\cdot, 0)$. The top-most panels depict three types of wavespeeds $c$. The left-most panels depict two realizations of initial pressures $q(\cdot, 0)$. The remaining panels display the final pressures $q(\cdot, T)$, which are influenced by both the initial pressure $q(\cdot, 0)$ in its respective row and the wavespeed $c$ in its corresponding column.

**Reverse time continuation (RTC) problem.** The goal of this problem is to determine the initial pressure of the wave equation (8) based on the final pressure. Let $q(\cdot, 0)$ be the initial pressure, which propagates over a time span $T$ to reach the final pressure $q(\cdot, T)$. Note that the final pressure $q(\cdot, T)$ is influenced by both the initial pressure $q(\cdot, 0)$ and the wavespeed $c$ in (8). With wavespeed $c$ fixed, the ground-truth operator that solves the RTC task is represented by

$$G_{\text{rtc}}^{\dagger} : q(\cdot, T) \mapsto q(\cdot, 0). \tag{9}$$

We construct the RTC dataset in the form of $\big(q_j(\cdot, T), q_j(\cdot, 0)\big)_j$, with a fixed propagation time $T$ but different wavespeed types. Figure 3 illustrates samples from these datasets. We consider three types of wavespeed $c$ (top panels of Figure 3): Gaussian lens and realizations of isotropic and anisotropic Gaussian random fields. For configuring the initial pressure $q(\cdot, 0)$, we use two approaches. In the first approach, we place boxes or thick lines of random sizes, orientations, and locations in the domain, following the approach in Kothari et al. (2020). Additionally, to evaluate the out-of-distribution (OOD) generalization performance of models, we construct test datasets using MNIST images (LeCun et al., 1998) as the initial pressure. The two bottom rows of Figure 3 display samples from both the in-distribution thick line initial pressure and the OOD MNIST pressure. Appendix A provides detailed configuration information about the RTC datasets.

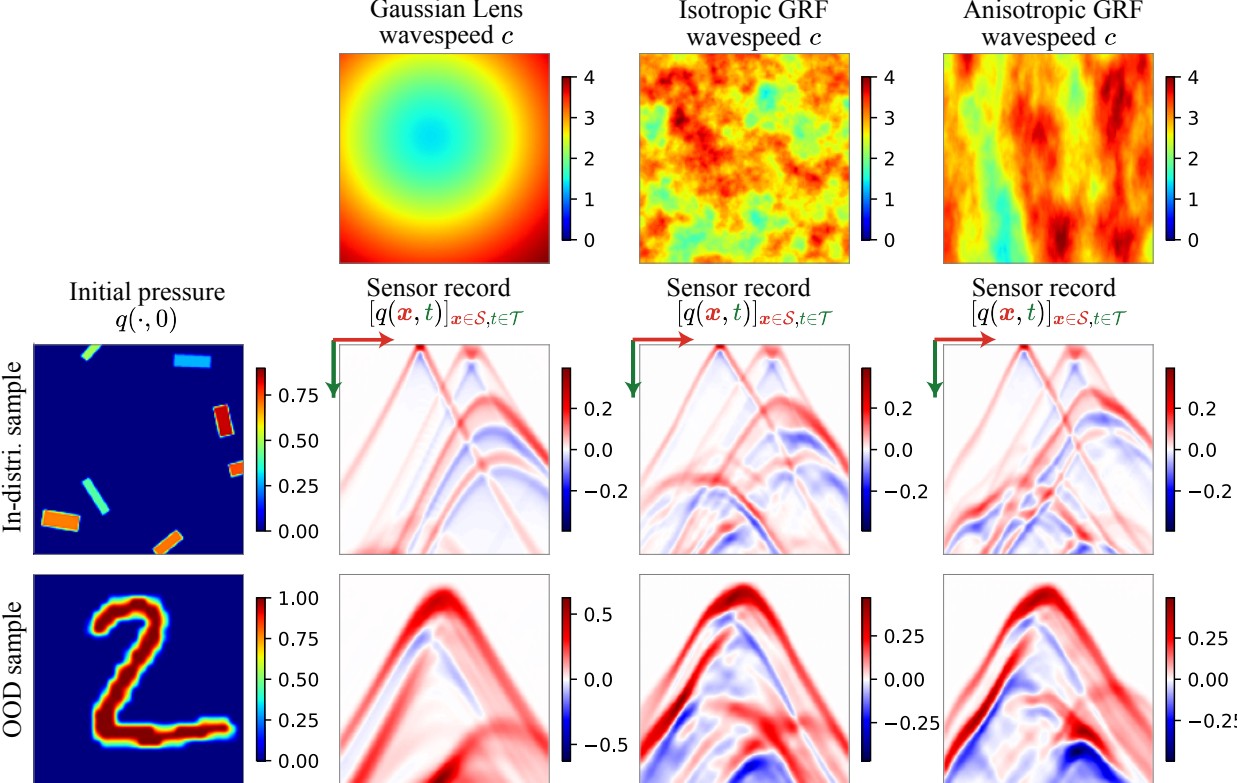

Figure 4: **Visualization of samples from the inverse source (IS) problems**. At a fixed wavespeed $c$, the goal of IS is to map $[q(\boldsymbol{x}, t)]_{\boldsymbol{x} \in \mathcal{S}, t \in \mathcal{T}}$ (pressure recorded at sensor locations $\mathcal{S}$ and time $\mathcal{T}$) into the initial pressure $q(\cdot, 0)$. The top-most panels depict three types of wavespeeds $c$. The left-most panels depict two realizations of initial pressures $q(\cdot, 0)$. The remaining panels exhibit the sensor records $[q(\boldsymbol{x}, t)]_{\boldsymbol{x} \in \mathcal{S}, t \in \mathcal{T}}$, which are influenced by both the initial pressure $q(\cdot, 0)$ in its respective row and the wavespeed $c$ in its corresponding column.

**Inverse source (IS) problem.** In the previous RTC problem, we are given the final pressure $q(\cdot, T)$ on the full domain at a terminal time $T$. However, this assumption is often too strong in applications such as seismic imaging, where the pressure can only be measured by sensors placed at some portions of the domain

boundary. To address this limitation, we turn to the inverse source (IS) problem. The IS problem aims to predict the initial pressure $q(\cdot, 0)$ based on pressure measurements taken at certain boundary locations over a time span $[0, T]$. Given a fixed wavespeed $c$, sensor locations $S$, and discrete time steps $\mathcal{T}$, the ground-truth operator that solves the IS task can be expressed as:

$$G_{\mathrm{is}}^{\dagger} : [q(\boldsymbol{x}, t)]_{\boldsymbol{x} \in \mathcal{S}, t \in \mathcal{T}} \mapsto q(\cdot, 0). \tag{10}$$

This problem is illustrated in Figure 4. Similar to the RTC dataset, we create random thick lines and use them as initial pressure $q(\cdot, 0)$. In Figure 4, we visualize samples from IS datasets. The sensor locations $\mathcal{S}$ are fixed to be the upper boundary of the domain. Further details on the configuration of the IS datasets are provided in Appendix A.

We summarize our 12 time-varying datasets of RTC and IS in Table 3 in Appendix A. Each dataset based upon the initial pressures of thick lines consists of 9,000 training samples, 500 validation samples, and 500 testing samples. Each dataset based upon the initial pressures of MNIST is for out-of-distribution testing purposes, comprising of 500 testing samples.

### 3.3  Dataset accessibility and format.

The datasets are in the `beton` format of FFCV (Leclerc et al., 2023), which is open-source software that provides high-throughput data loading for model training. Our datasets are accessible on Zenodo (an open platform for datasets sharing): `https://zenodo.org/records/8015145`, and the benchmark code is accessible through our GitHub repository: `https://github.com/wavebench/wavebench`.

## 4  Wavebench Benchmarks

### 4.1  Baseline models

We provide reference implementations and benchmark performance of PDE surrogates trained on WAVEBENCH datasets. We focus on U-Nets and FNOs as PDE surrogates, which exhibit high performance across various PDE problems (Takamoto et al., 2022; Gupta & Brandstetter, 2022).

| Model | # parameters | Forward pass runtime [s] | Backward pass runtime [s] |
|---|---|---|---|
| FNO-depth-4 | 4.2M | 0.011 | 0.018 |
| FNO-depth-8 | 8.4M | 0.019 | 0.033 |
| U-Net-ch-32 | 7.8M | 0.006 | 0.012 |
| U-Net-ch-64 | 31.0M | 0.016 | 0.031 |
| UNO-modes-12 | 10.1M | 0.024 | 0.032 |
| UNO-modes-16 | 17.9M | 0.024 | 0.033 |

Table 2: **Comparison of baseline models.** The baseline models include two variants of FNO (Li et al., 2021) and two variants of U-Net (Ronneberger et al., 2015). FNO-depth-4 and FNO-depth-8 are two FNO variants with of 4 or 8 hidden Fourier layers. U-Net-ch-64 stands for the standard U-Net that has 64 channels in its first layer, and U-Net-ch-32 is a smaller variant with all convolutional channels halved. Both UNO-modes-12 and UNO-modes-16 variants have 3 scales that correspond to domain-discretization of $128 \times 128$, $64 \times 64$, and $32 \times 32$. The UNO-modes-12 model uses $[12, 6, 3]$ Fourier modes for each scale, while the UNO-modes-16 uses $[16, 8, 4]$ Fourier modes for each scale. The model runtime was assessed using a batch $(8, 1, 128, 128)$, consisting of 8 samples of $128 \times 128$ array. The benchmarking procedure involved initial 10 dry runs followed by 100 test runs conducted on an 11 GB NVIDIA GeForce RTX 2080 Ti GPU.

**FNO.**  The Fourier neural operator (Li et al., 2021) represents one of the state-of-the-art models for PDE data on regular grids. We use FNOs in 2D consisting of 64 hidden channels, 16 Fourier modes, and either 4 or 8 Fourier layers as hidden layers; for the input lifting and output projection parts of FNOs, we use

single-hidden-layer MLPs that consisted of 1x1 convolution with 128 channels. The FNO variants with 4 and 8 hidden Fourier layers are referred to as FNO-depth-4 and FNO-depth-8, whose parameter count and runtime are summarized in Table 2.

**U-Net.** The U-Net (Ronneberger et al., 2015) is a convolutional network originally developed for 2D image-to-image regression problems such as image segmentation. However, its versatility has led to its adoption in PDE learning tasks (Kothari et al., 2020; Li et al., 2021; Gupta & Brandstetter, 2022; Chen & Thuerey, 2023), both as a component within larger architectures or as standalone model. Remarkably, even as a standalone model, U-Nets perform well in several PDE modeling tasks, sometimes matching or surpassing dedicated PDE surrogates such as FNOs (Takamoto et al., 2022; Gupta & Brandstetter, 2022). In our baselines, we employ either the standard U-Net (Ronneberger et al., 2015), referred to as U-Net-ch-64 as it uses 64 channels in the first hidden layer, or a smaller variant with halved convolutional channels (referred to as U-Net-ch-32). Table 2 summarizes their parameter counts and runtime.

**UNO.** The UNO (U-shaped neural operator) (Rahman et al., 2023) combines U-Net and FNO design elements. Like U-Net, UNO encodes input into smaller domains and decodes to generate output on greater domains. UNO's layers are parametrized with Fourier layers in a way identical to FNOs. We consider two UNO variants, UNO-modes-12 and UNO-modes-16. For both versions, we use 3 scales on domains $128 \times 128$, $64 \times 64$, and $32 \times 32$. UNO-modes-12 uses Fourier modes of $[12, 6, 3]$ for each scale, and UNO-modes-16 with $[16, 8, 4]$ for each scale.

**Training protocol.** We trained and tested the baseline U-Net, FNO, and UNO models using the 20 datasets described in Section 3 and summarized in Table 1 and Table 3 in the appendix. For all datasets, we trained all models for 50 epochs using the AdamW optimizer (Loshchilov & Hutter, 2019). The learning rates were initially set to 1e-3 and then annealed to 1e-5 using the cosine annealing (Loshchilov & Hutter, 2017). We employed the relative L2 loss for training and evaluation in all our problems, following the approach in Li et al. (2021); de Hoop et al. (2022b). Since all datasets were divided into training, validation, and test splits, we monitored the model's generalization performance on the validation split during training. We selected the model that performed the best on the validation split for the evaluation of the test split. All experiments were conducted on an 11 GB NVIDIA GeForce RTX 2080 Ti GPU.

In the following sections, we present empirical results on a subset of WAVEBENCH datasets. Additional visualizations and complete qualitative results are provided in Appendix A. Due to space limitations, we present results for PDE surrogates of FNOs and U-Nets here in the main text; UNO variant outcomes are in the appendix. In most experiments, UNOs outperform some U-Nets but not the best FNOs.

### 4.2 Performance of baseline models on time-harmonic datasets

**In-distribution performance.** Figure 5 illustrates the performances of models applied to the time-harmonic dataset with isotropic GRF wavespeeds and frequency 40 Hz. All models are trained using the training split of the dataset; the figure presents results based on a sample from the test split. All PDE surrogates produce visually comparable predictions. However, the error fields (the third row of Figure 5) show that two FNO variants produce smaller errors in comparison to U-Nets.

Numerical values of the errors are provided in Table 4 in the Appendix B.

**Out-of-distribution (OOD) performance.** While FNOs and U-Nets produce both visually and numerically appealing results for in-distribution samples, their out-of-distribution (OOD) test results clearly indicate room for further improvements (Figure 6). In this experiment, we trained models on a time-harmonic dataset with *isotropic* GRF wavespeed and evaluated them on the *anisotropic* version of the dataset. This setup presents a greater challenge, as models that can successfully solve it presumably must capture a greater extent of the physics of wave propagation. The results demonstrate that PDE surrogates generally perform over 5 times worse in OOD tests compared to in-distribution tests in the metric of relative L2 error (Table 5 in the Appendix B).

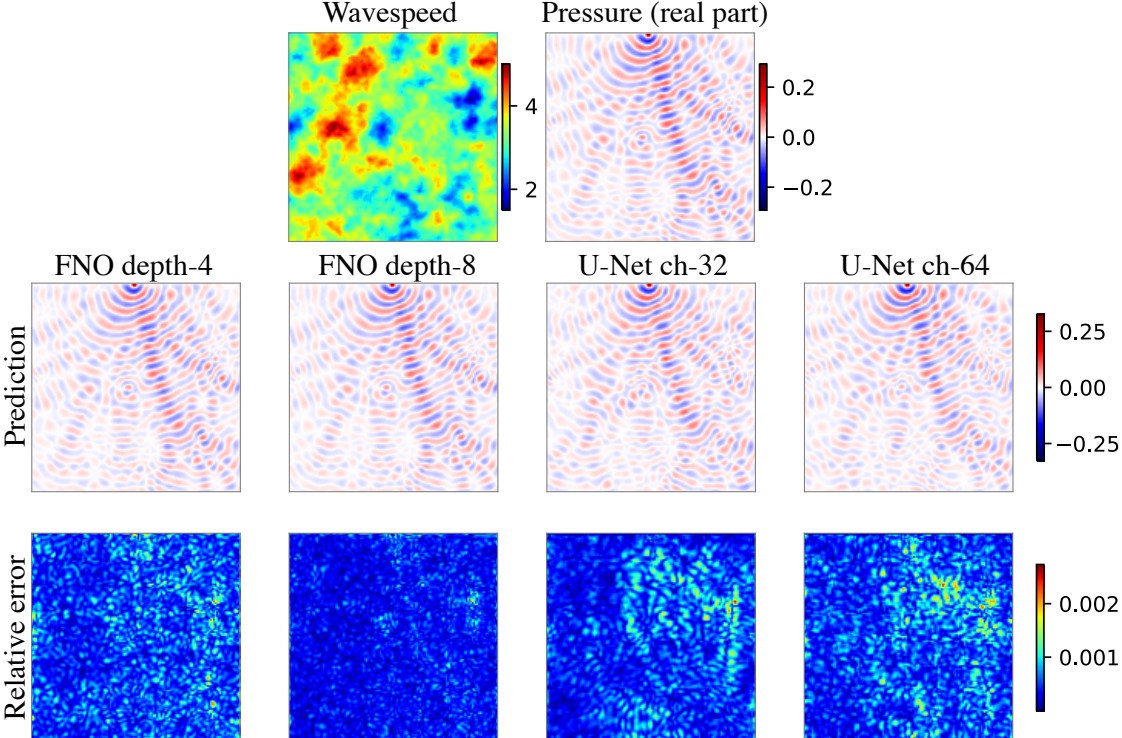

Figure 5: **In-distribution test performance of models on a acoustic time-harmonic dataset.** The model is trained and tested on splits of the acoustic time-harmonic dataset with isotropic GRF wavespeed and frequency $\omega/2\pi = 40$Hz. The input wavespeed $c$ and real part of the ground-truth pressure field are shown in the top panels, cf. Equation (1). The middle panels show the predictions from 4 different models. The bottom panels show the relative error $|p(\cdot,\omega) - \widehat{p}(\cdot,\omega)| / \|p(\cdot,\omega)\|_{L^2}$ between the predicted pressure $\widehat{p}$ and the ground-truth pressure $p$.

### 4.3 Performance of baseline models on time-harmonic datasets

Recall that WAVEBENCH contains two time-varying wave problems: reverse time continuation (RTC) and the inverse source problem (IS). Here in the main text, we report the model performances on the more challenging IS problem. Full results of both problems can be found in Appendix D.

Figure 7 provides a visual performances comparison of PDE surrogates on the IS dataset that uses isotropic Gaussian Random Field (GRF) wavespeeds. Both U-Nets and Fourier neural operators yield comparable results on in-distribution samples. U-Nets slightly exceed FNOs in performance, though the margin is narrow (Table 7 in Appendix D). The outputs from both models are within reasonable expectations.

However, when it comes to OOD samples, both models introduce artifacts, leading to inaccuracies in reconstructing the target. This can be seen in the third row of Figure 7. There, U-Nets generate outputs with box-like patterns, which are derived from the training data; FNOs struggle to accurately reproduce the smooth contour of the OOD target. It is worth noting that similar artifacts can be consistently observed across all time-varying datasets (Appendix D).

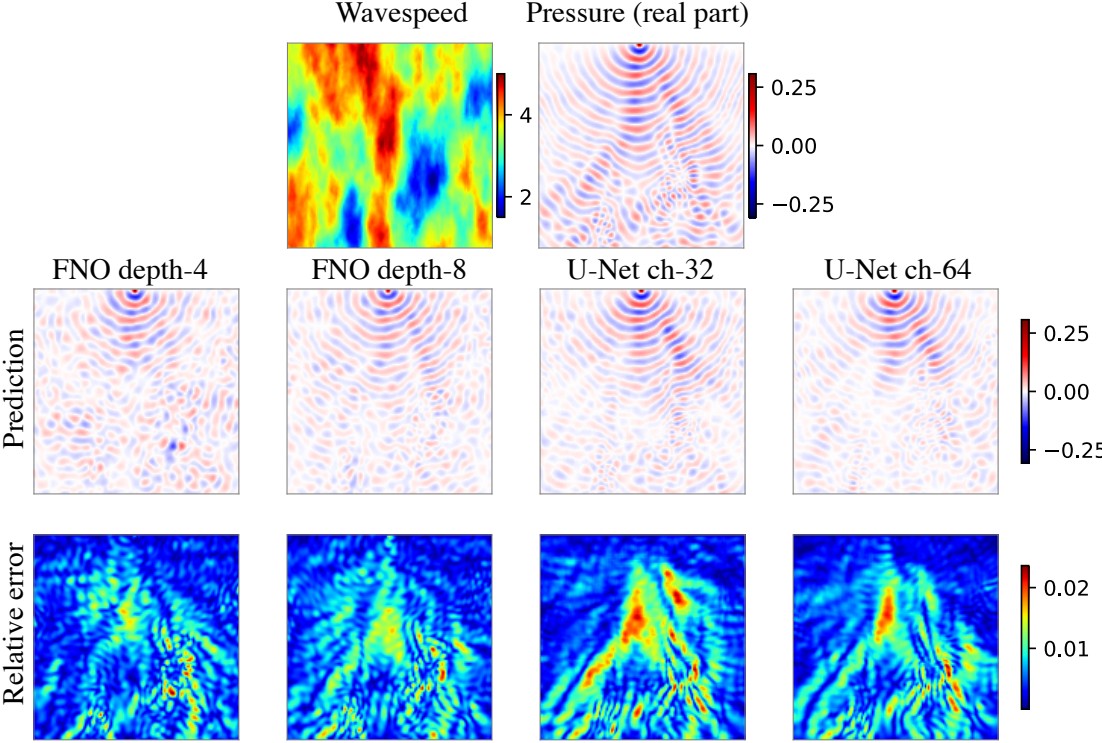

Figure 6: **OOD test performance of models on an acoustic time-harmonic dataset.** Here, the model is trained on the time-harmonic dataset with *isotropic* GRF wavespeed and frequency $\omega/2\pi = 40$Hz, but tested on the *anisotropic* version instead.

## 5   Discussion

This paper introduces WAVEBENCH, a comprehensive repository of 24 benchmark datasets designed for wave PDEs. The datasets cover a broad range of wave-related problems, including time-dependent problems derived from the wave equation and time-harmonic problems derived from the Helmholtz equation. WAVEBENCH serves two purposes: providing a data source for machine learning for wave problems and providing a user-friendly PyTorch environment for training and evaluating PDE surrogate models.

After evaluating PDE surrogates on the WAVEBENCH datasets, we have identified several recurring patterns. Firstly, across all datasets, U-Nets and FNOs showed remarkable capabilities in approximating wave propagation for in-distribution data. The larger models, such as FNO-depth-8 and U-Net-ch-64, outperformed their smaller counterparts like FNO-depth-4 and U-Net-ch-32. This shows the potential advantages of employing high-capacity models to tackle challenging PDE problems, assuming the existence of abundant and high-quality PDE data.

However, the performance of PDE surrogates declines significantly when moving from in-distribution to out-of-distribution (OOD) samples. This implies that although these models learn statistical patterns from the data, they do not capture the wave-propagation physics present in the data. Given the importance of OOD generalization in the practical use of wave-based techniques in exploratory scientific applications like seismic imaging and medical imaging, future PDE surrogates that have the capability to learn wave-propagation physics from data are highly sought-after.

**Limitations.** While WAVEBENCH contains a broad range of wave propagation PDE problems, we note that all WAVEBENCH datasets are presently simulated on 2D domains. A simulation of wave propagation within a 3D domain could provide a more realistic and challenging setting for applying PDE surrogates. At

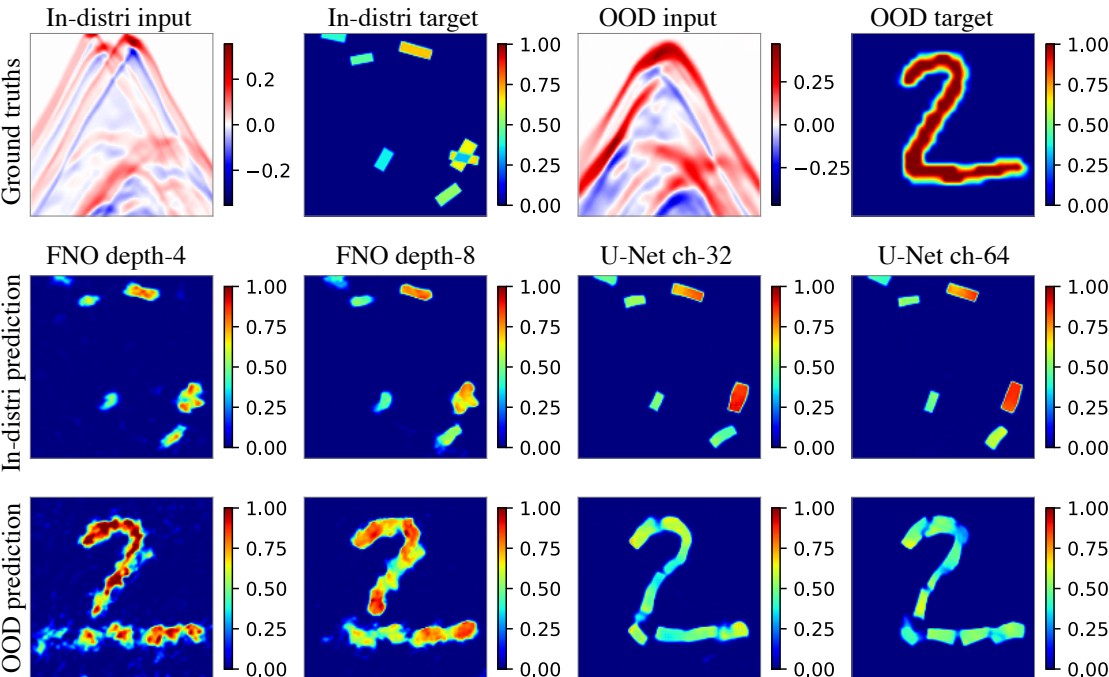

Figure 7: **Test performance of models on the time-varying IS dataset with isotropic GRF wavespeed.** The first row shows the input and target samples of the IS dataset; they can be either in-distribution or OOD. The second row shows the model predictions on the in-distribution sample. The third row shows the model predictions on the OOD sample.

the same time, preparing such a dataset is computationally much more demanding. We leave this aspect for future work.

## Acknowledgement.

This work was supported by the European Research Council Starting Grant 852821—SWING. Numerical experiments were partly performed at the sciCORE (http://scicore.unibas.ch/) scientific computing center at the University of Basel. FF acknowledges funding by the European Research Council Starting with Grant 101116288 – INCORWAVE. Views and opinions expressed are however those of the authors only and do not necessarily reflect those of the European Union or the European Research Council Executive Agency (ERCEA). Neither the European Union nor the granting authority can be held responsible for them.

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

# WaveBench: Benchmark Datasets for
# Modeling Linear Wave Propagation PDEs
# Supplementary Material

## A   Details of the time-varying datasets

To simulate wave propagation for both Reverse Time Continuation (RTC) and Inverse Scattering (IS) problems, we use the open-source `j-wave package` (Stanziola et al., 2023). The j-wave package simulates the wave dynamics in (8) by an equivalent system of first-order equations (Treeby et al., 2012; Pierce, 2019):

$$\frac{\partial u}{\partial t} = -\frac{1}{b_0}\nabla q, \qquad \text{(momentum conservation)}$$

$$\frac{\partial b}{\partial t} = -b_0 \nabla \cdot u, \qquad \text{(mass conservation)}$$

$$q = c^2 b. \qquad \text{(pressure-density relation)}$$

where $u = u(\boldsymbol{x}, t)$ is called the acoustic particle velocity and $b_0$ is ambient density. Radiating boundary conditions are enforced with a perfectly matched layer (PML), following the default setting of `j-wave` (Stanziola et al., 2023).

In our simulation, the domain is represented as a square grid, with dimensions of 1.024 km $\times$ 1.024 km discretized into a $128 \times 128$ array. Recall that for the time-varying experiments, the wavespeed $c$ can be a realization of an isotropic GRF, anisotropic GRF, or a Gaussian lens. In the case of isotropic and anisotropic GRF, the wavespeeds are taken from the time-harmonic datasets; in the case of the Gaussian-lens wavespeed, the wavespeed is a point mass situated at the grid coordinates of $(50, 55)$ blurred by a Gaussian filter with a standard deviation 50 in both spatial directions. Across all types of wavespeeds considered in our simulations, the minimum wavespeed is normalized to a value of $1.4 \text{ km s}^{-1}$, while the maximum wavespeed is normalized to $4 \text{ km s}^{-1}$. The propagation time for both the RTC and IS simulations is set to be $T = 0.2$ s.

For both RTC and IS datasets, the initial pressure datasets $q(\cdot, 0)$ can either be thick lines or MNIST images, represented by $128 \times 128$ arrays; see Figure 3 and Figure 4. The thick lines represent pressures that are used as in-distribution samples for training and evaluating the model. They are box-like patterns of random sizes, orientations, and locations in the domain, following the approach in Kothari et al. (2020). Each sample contains 5 to 10 boxes, uniformly distributed. The box centroids are sampled on the discretized grid of the domain. Dimensions of boxes are sampled from uniform distributions: length from $[50, 100]$, width from $[20, 40]$, and orientation from $[0, \pi]$. The dataset consists of 9000 training samples, 500 validation samples, and 500 test samples. Additionally, there are 500 out-of-distribution (OOD) MNIST pressure samples for testing.

The IS problem is more challenging than RTC. This is because in IS we only get to measure the wave pressure at the top of the domain. That is, the sensor locations $\mathcal{S}$ in the measurements $[q(\boldsymbol{x}, t)]_{\boldsymbol{x} \in \mathcal{S}, t \in \mathcal{T}}$ correspond to the topmost coordinates of the domain (excluding the size of PML). The time steps $\mathcal{T}$ consist of 128 equidistant intervals within the range of $[0, T]$. These settings result in the sensor record $[q(\boldsymbol{x}, t)]_{\boldsymbol{x} \in \mathcal{S}, t \in \mathcal{T}}$ having a square image-like appearance as in Figure 4. To make the IS problem more tractable to solve, we use the following way to prepare the initial pressure $q(\cdot, 0)$. We resize thick line and MNIST images that with an original size $128 \times 128$ into the size of $64 \times 64$ and put them on the top center of the domain. The remaining entries are filled with zeros. Consequently, all nonzero entries of the initial wave pressure $q(\cdot, T)$ are concentrated in the top center of the domain. This configuration, with the sensors positioned at the top, allows for better reception of propagated waves and helps mitigate the ill-posedness of the problem.

| Problem | Wavespeed $c$ | Initial pressure $q(\cdot, 0)$ |
|---|---|---|
| Reverse time continuation (RTC) | Gaussian lens | Thick lines MNIST |
| | Isotropic GRF | Thick lines MNIST |
| | Anisotropic GRF | Thick lines MNIST |
| Inverse source (IS) | Gaussian lens | Thick lines MNIST |
| | Isotropic GRF | Thick lines MNIST |
| | Anisotropic GRF | Thick lines MNIST |

Table 3: **Summary of the 12 time-varying datasets.** Each dataset corresponds to specific problem types (reverse time continuation or inverse source), wavespeed variations (Gaussian lens, isotropic GRF, or anisotropic GRF), and initial pressure characteristics (thick lines or MNIST). The thick line initial pressure datasets consist of in-distribution samples: they contain **9000 training samples, 500 validation samples, and 500 testing samples**. The MNIST pressure dataset consists of out-of-distribution (OOD) samples used exclusively for testing and comprises **500 samples**.

## B  Full experimental results of the acoustic time-harmonic datasets

| Wavespeed $c$ | Freq. $\omega/2\pi$ | FNO-depth-4 | FNO-depth-8 | U-Net-ch-32 | U-Net-ch-64 | UNO-modes-12 | UNO-modes-16 |
|---|---|---|---|---|---|---|---|
| Isotropic GRF | 10 Hz | 0.063 | 0.040 | 0.073 | 0.063 | 0.064 | 0.054 |
| | 15 Hz | 0.093 | 0.057 | 0.116 | 0.087 | 0.106 | 0.081 |
| | 20 Hz | 0.122 | 0.070 | 0.157 | 0.106 | 0.147 | 0.114 |
| | 40 Hz | 0.283 | 0.165 | 0.286 | 0.191 | 0.407 | 0.301 |
| Anisotropic GRF | 10 Hz | 0.059 | 0.025 | 0.144 | 0.119 | 0.074 | 0.051 |
| | 15 Hz | 0.098 | 0.039 | 0.204 | 0.165 | 0.123 | 0.093 |
| | 20 Hz | 0.135 | 0.060 | 0.230 | 0.176 | 0.171 | 0.129 |
| | 40 Hz | 0.315 | 0.172 | 0.321 | 0.231 | 0.422 | 0.343 |

Table 4:   In-distribution performance comparison of models on the test folds of the time-harmonic datasets.  The error metric is the relative L2 error $\|p - \widehat{p}\|_{L^2} / \|p\|_{L^2}$ between the ground-truth $p$ and prediction $\widehat{p}$.

| Wavespeed $c$ | Freq. $\omega/2\pi$ | FNO-depth-4 | FNO-depth-8 | U-Net-ch-32 | U-Net-ch-64 | UNO-modes-12 | UNO-modes-16 |
|---|---|---|---|---|---|---|---|
| Isotropic GRF | 10 Hz | 0.485 | 0.379 | 0.527 | 0.506 | 0.489 | 0.458 |
| | 15 Hz | 0.633 | 0.464 | 0.638 | 0.620 | 0.674 | 0.618 |
| | 20 Hz | 0.758 | 0.533 | 0.751 | 0.717 | 0.770 | 0.747 |
| | 40 Hz | 1.152 | 0.895 | 0.883 | 0.891 | 0.893 | 0.951 |
| Anisotropic GRF | 10 Hz | 0.560 | 0.388 | 0.541 | 0.527 | 0.376 | 0.382 |
| | 15 Hz | 0.771 | 0.518 | 0.671 | 0.656 | 0.498 | 0.483 |
| | 20 Hz | 0.812 | 0.612 | 0.754 | 0.725 | 0.607 | 0.599 |
| | 40 Hz | 1.018 | 0.887 | 0.905 | 0.898 | 0.803 | 0.950 |

Table 5:   OOD performance comparison of models on the test folds of the time-harmonic datasets.  The table layout is similar to Table 4.  The wavespeed $c$ reported in the left column shows the OOD wavespeed used in test data; for instance, the three "Isotropic GRF" rows are based on models trained on the corresponding Anisotropic GRF versions, and vice versa.

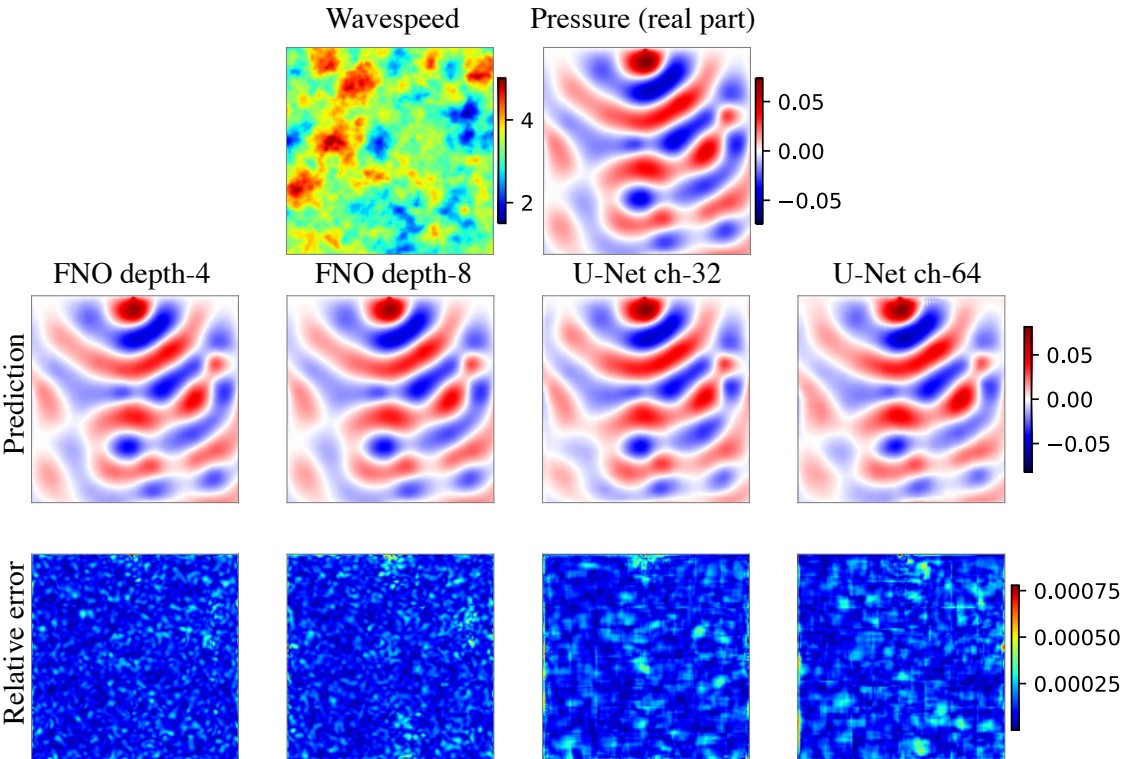

Figure 8: In-distribution test performance of models on an acoustic time-harmonic dataset. The time-harmonic dataset is configured with isotropic GRF wavespeed and frequency $\omega/2\pi = 10$Hz. The figure layout is same as Figure 5 in the main text.

## C Full experimental results of the elastic time-harmonic datasets

| Freq. $\omega/2\pi$ | FNO-depth-4 | FNO-depth-8 | U-Net-ch-32 | U-Net-ch-64 | UNO-modes-12 | UNO-modes-16 |
|---|---|---|---|---|---|---|
| 10 Hz | 0.080 | 0.039 | 0.154 | 0.141 | 0.103 | 0.076 |
| 15 Hz | 0.130 | 0.072 | 0.229 | 0.217 | 0.205 | 0.127 |
| 20 Hz | 0.225 | 0.124 | 0.267 | 0.225 | 0.264 | 0.204 |
| 40 Hz | 0.504 | 0.365 | 0.497 | 0.470 | 0.534 | 0.490 |

Table 6: In-distribution performance comparison of models on the test folds of the elastic time-harmonic datasets. Wavespeeds are anisotropic GRFs for all frequencies. The error metric is the relative L2 error $\|\boldsymbol{u} - \widehat{\boldsymbol{u}}\|_{L^2}/\|\boldsymbol{u}\|_{L^2}$ between the ground-truth $\boldsymbol{u}$ and prediction $\widehat{\boldsymbol{u}}$.

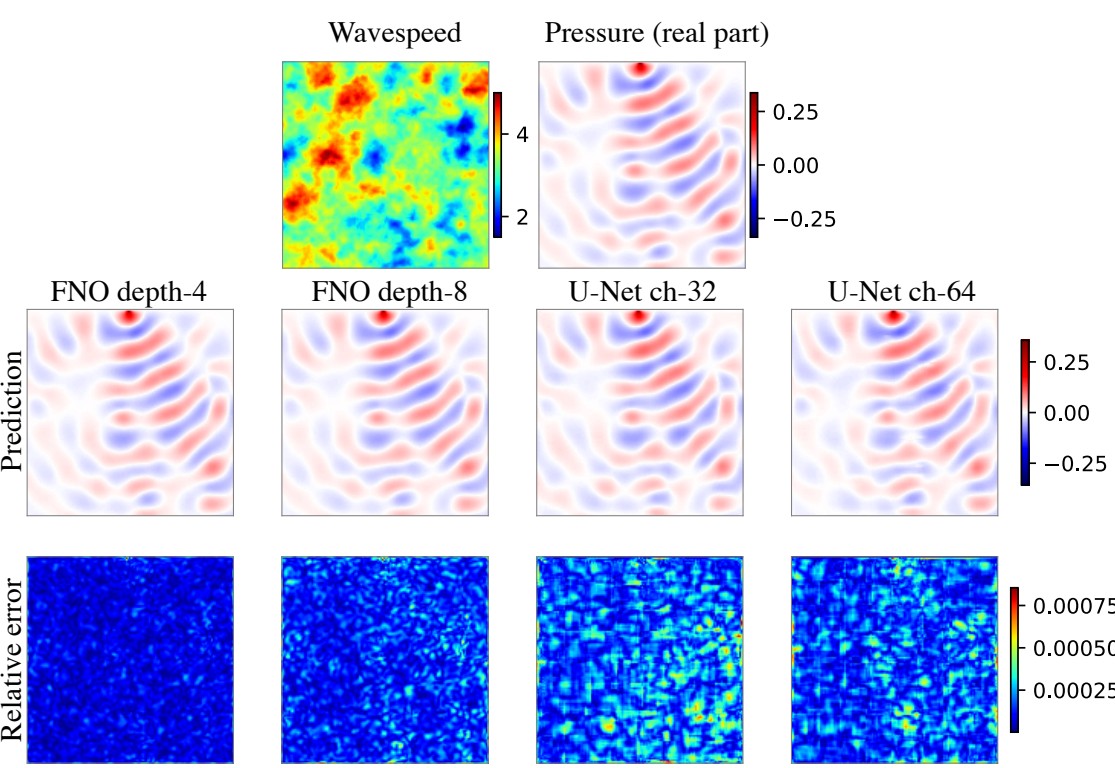

Figure 9: In-distribution test performance of models on a acoustic time-harmonic dataset. The time-harmonic dataset is configured with with isotropic GRF wavespeed and frequency $\omega/2\pi = 15$Hz. The figure layout is same as Figure 5 in the main text.

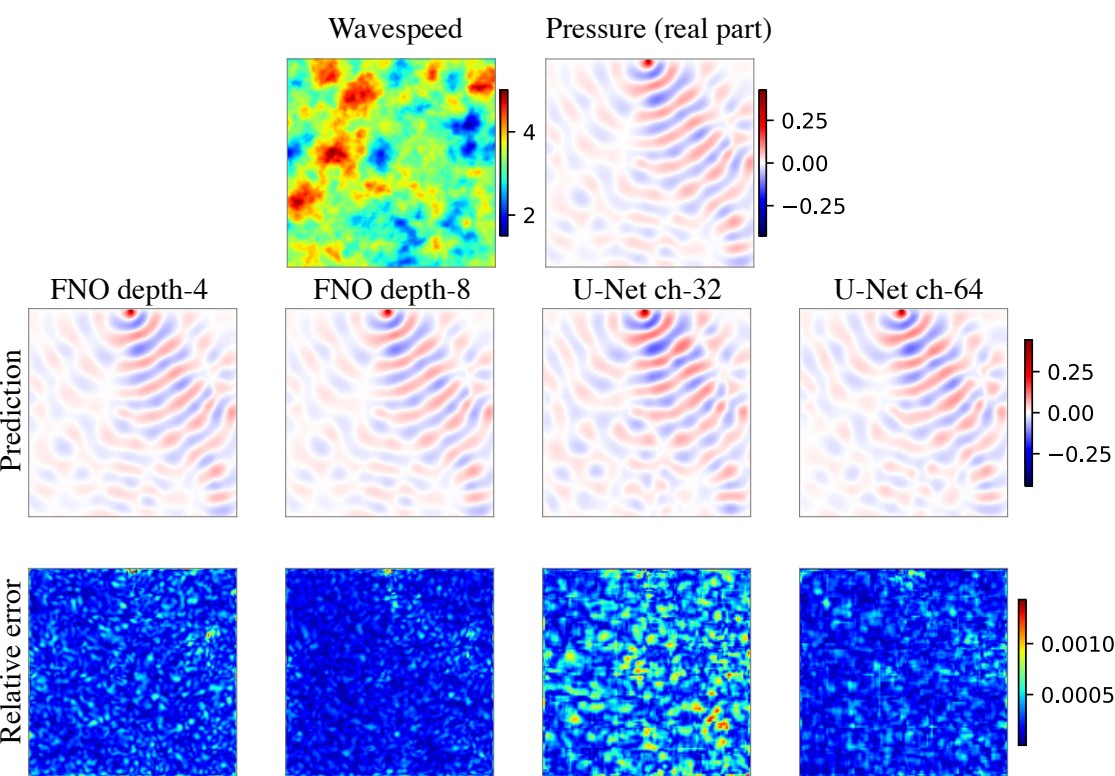

Figure 10: In-distribution test performance of models on a acoustic time-harmonic dataset. The time-harmonic dataset is configured with isotropic GRF wavespeed and frequency $\omega/2\pi = 20$Hz. The figure layout is same as Figure 5 in the main text.

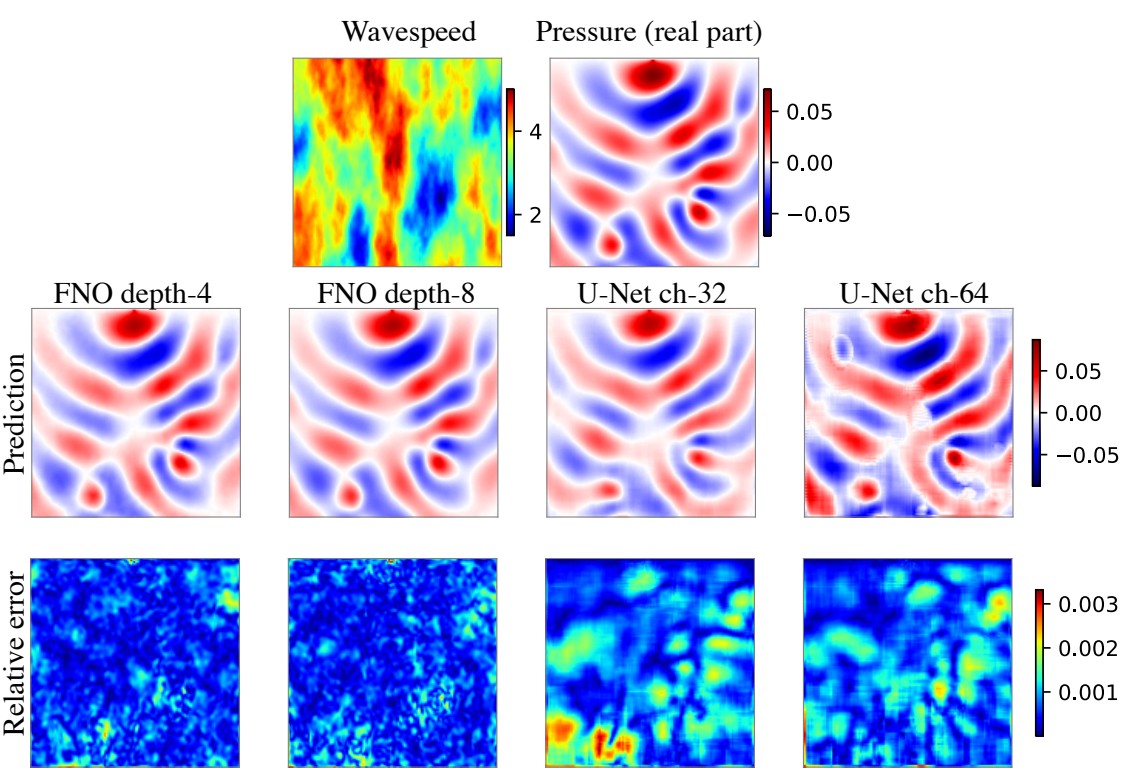

Figure 11: In-distribution test performance of models on a acoustic time-harmonic dataset. The time-harmonic dataset is configured with anisotropic GRF wavespeed and frequency $\omega/2\pi = 10$Hz. The figure layout is same as Figure 5 in the main text.

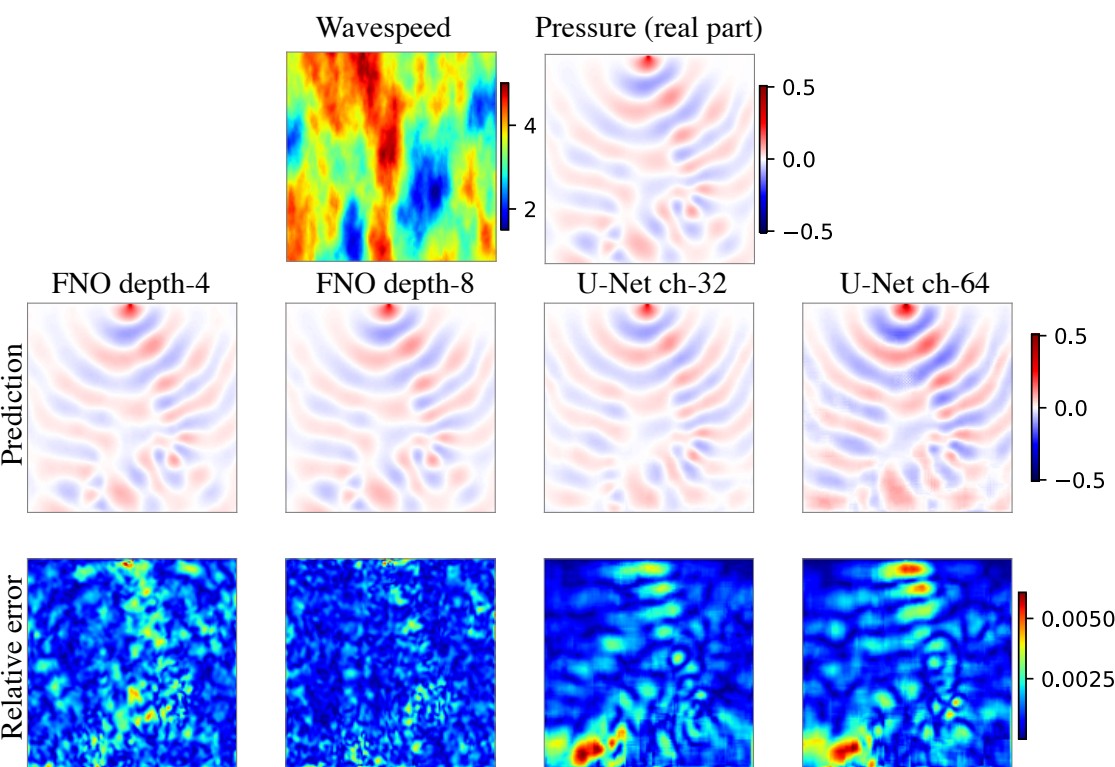

Figure 12: In-distribution test performance of models on a acoustic time-harmonic dataset. The time-harmonic dataset is configured with anisotropic GRF wavespeed and frequency $\omega/2\pi = 15$Hz. The figure layout is same as Figure 5 in the main text.

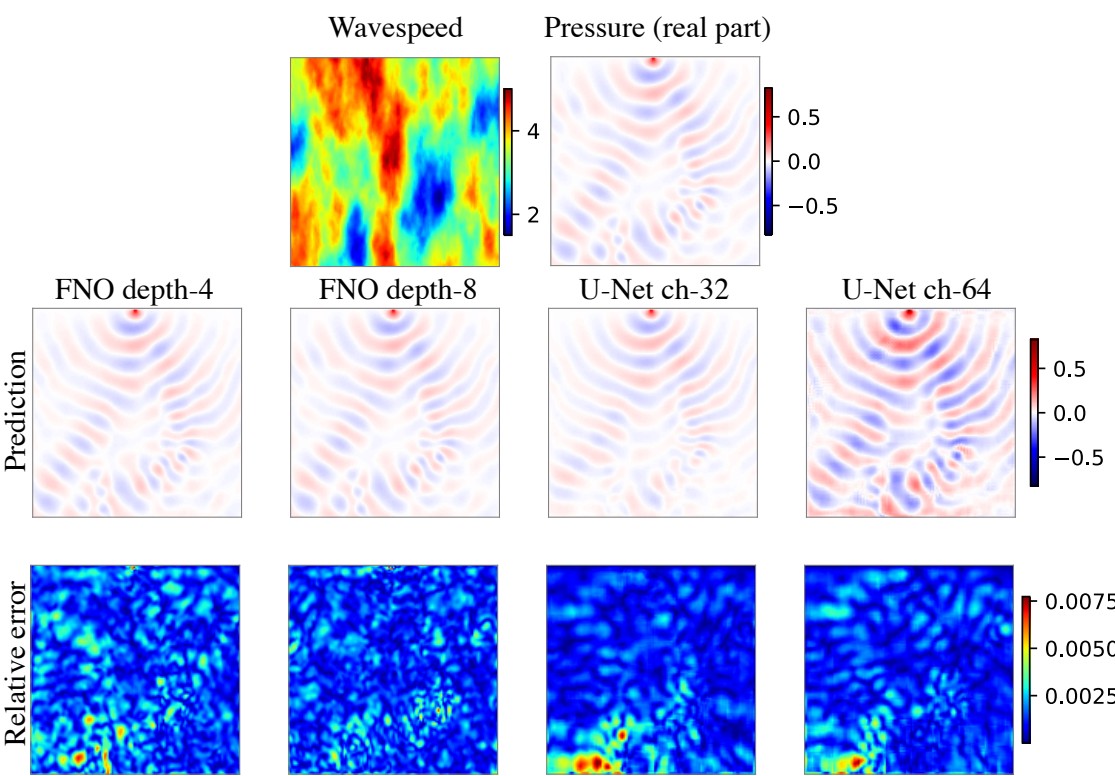

Figure 13: In-distribution test performance of models on a acoustic time-harmonic dataset. The time-harmonic dataset is configured with anisotropic GRF wavespeed and frequency $\omega/2\pi = 20$Hz. The figure layout is same as Figure 5 in the main text.

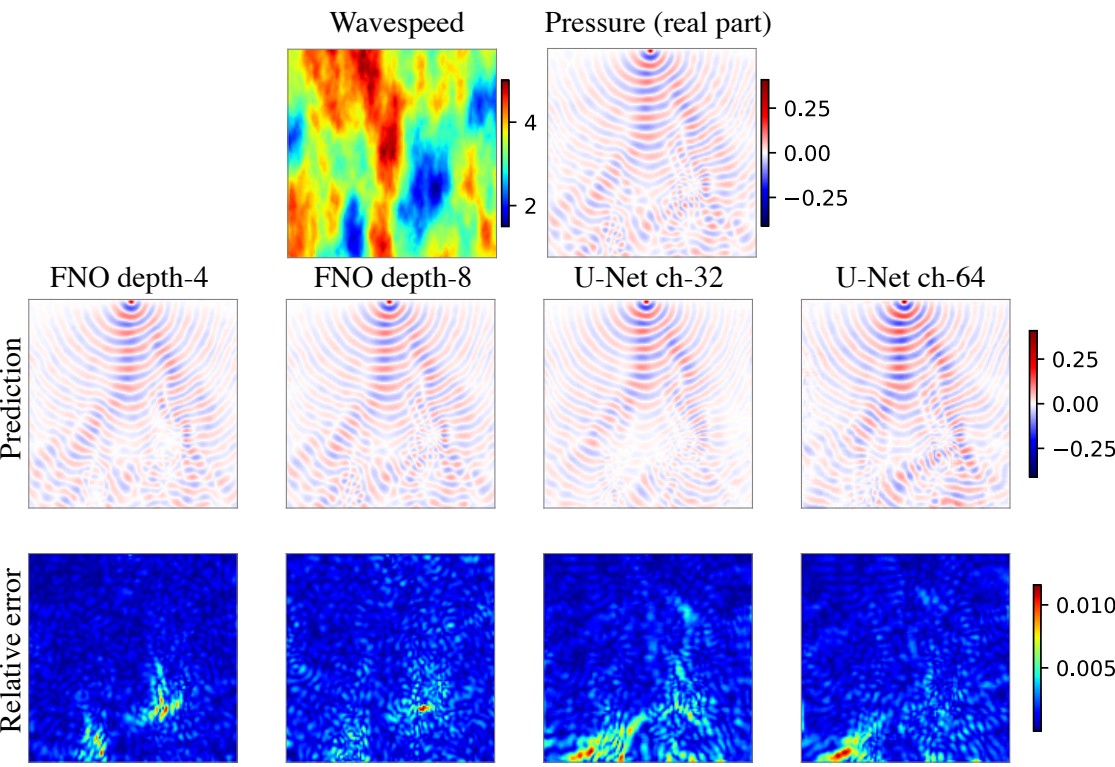

Figure 14: In-distribution test performance of models on a acoustic time-harmonic dataset. The time-harmonic dataset is configured with anisotropic GRF wavespeed and frequency $\omega/2\pi = 40$Hz. The figure layout is same as Figure 5 in the main text.

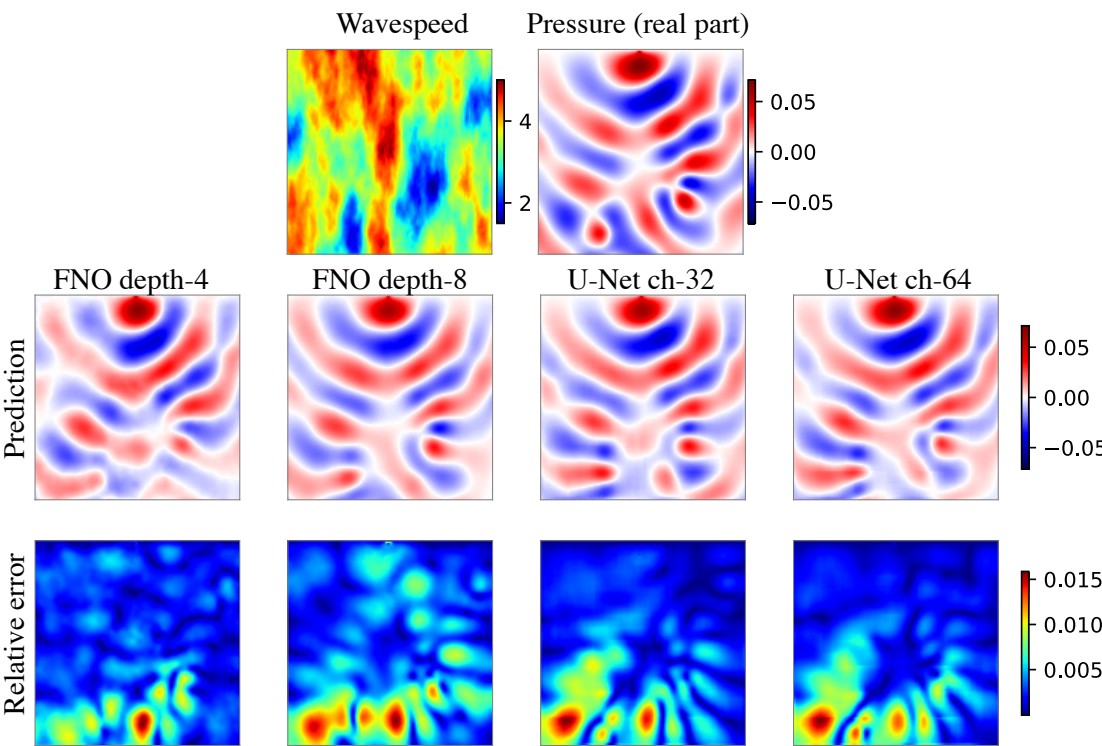

Figure 15: OOD test performance of models on a time-harmonic dataset. The model is trained on the time-harmonic dataset with *isotropic* GRF wavespeed and frequency $\omega/2\pi = 10$Hz, but tested on the *anisotropic* version instead. The figure layout follows from Figure 6 in the main text.

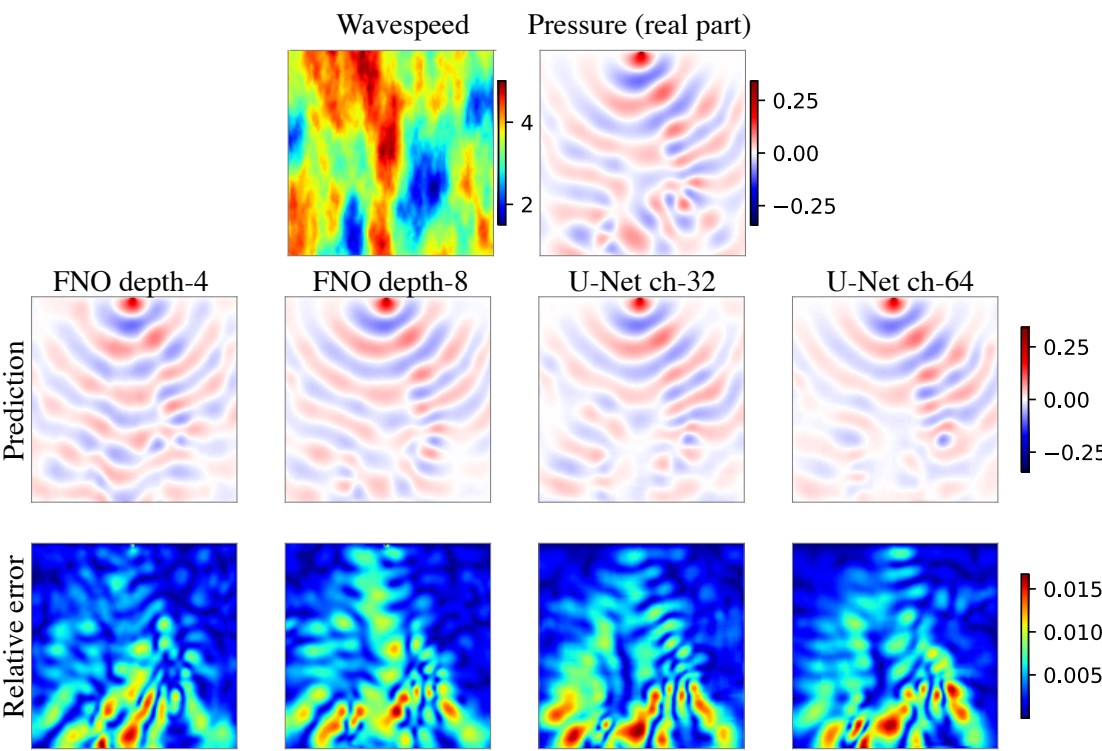

Figure 16: OOD test performance of models on a time-harmonic dataset. The model is trained on the time-harmonic dataset with *isotropic* GRF wavespeed and frequency $\omega/2\pi = 15$Hz, but tested on the *anisotropic* version instead. The figure layout follows from Figure 6 in the main text.

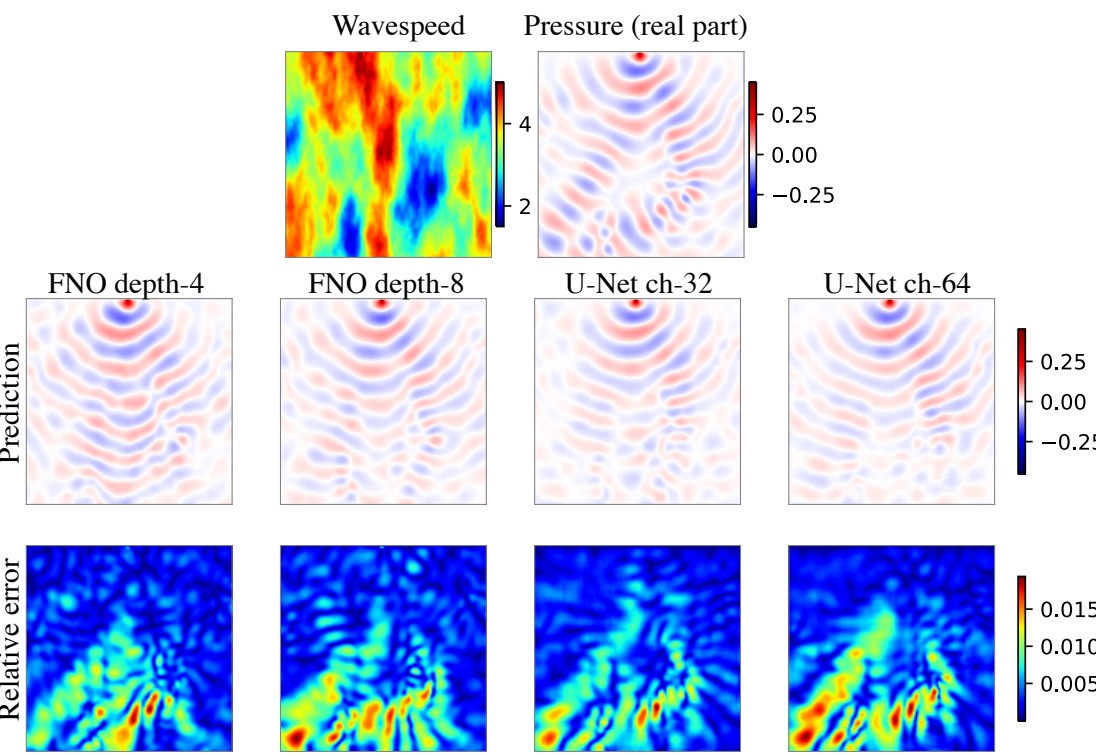

Figure 17: OOD test performance of models on a time-harmonic dataset. The model is trained on the time-harmonic dataset with *isotropic* GRF wavespeed and frequency $\omega/2\pi = 20$Hz, but tested on the *anisotropic* version instead. The figure layout follows from Figure 6 in the main text.

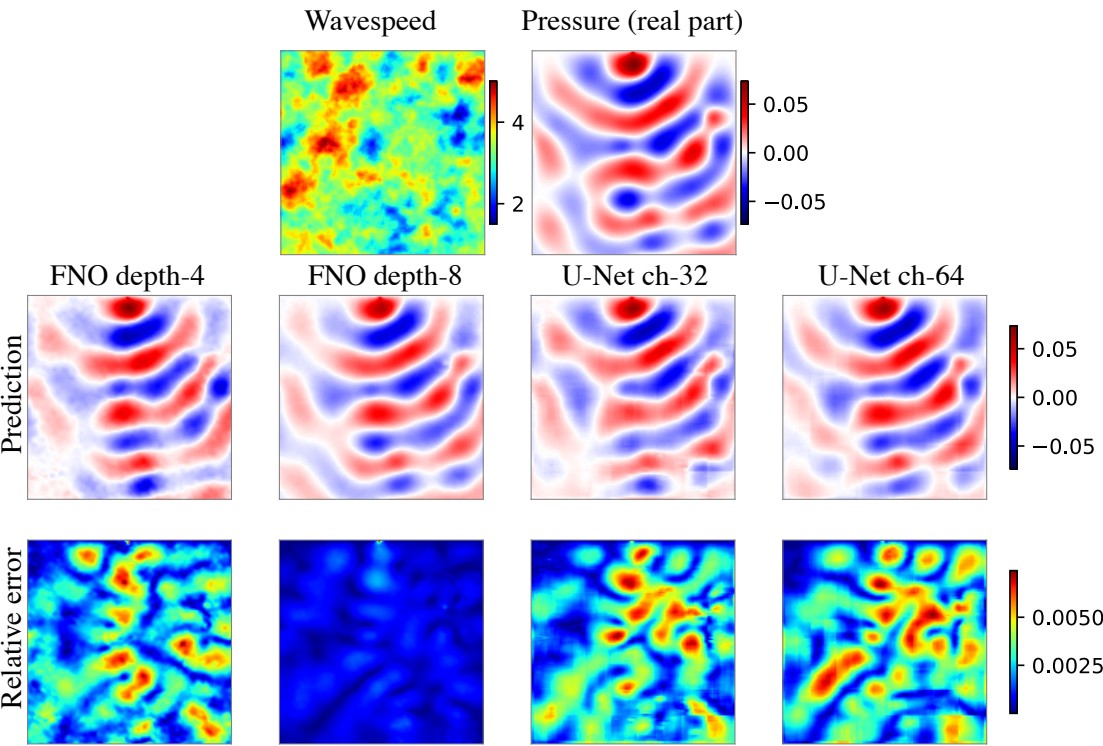

Figure 18: OOD test performance of models on a time-harmonic dataset. The model is trained on the time-harmonic dataset with *isotropic* GRF wavespeed and frequency $\omega/2\pi = 10$Hz, but tested on the *anisotropic* version instead. The figure layout follows from Figure 6 in the main text.

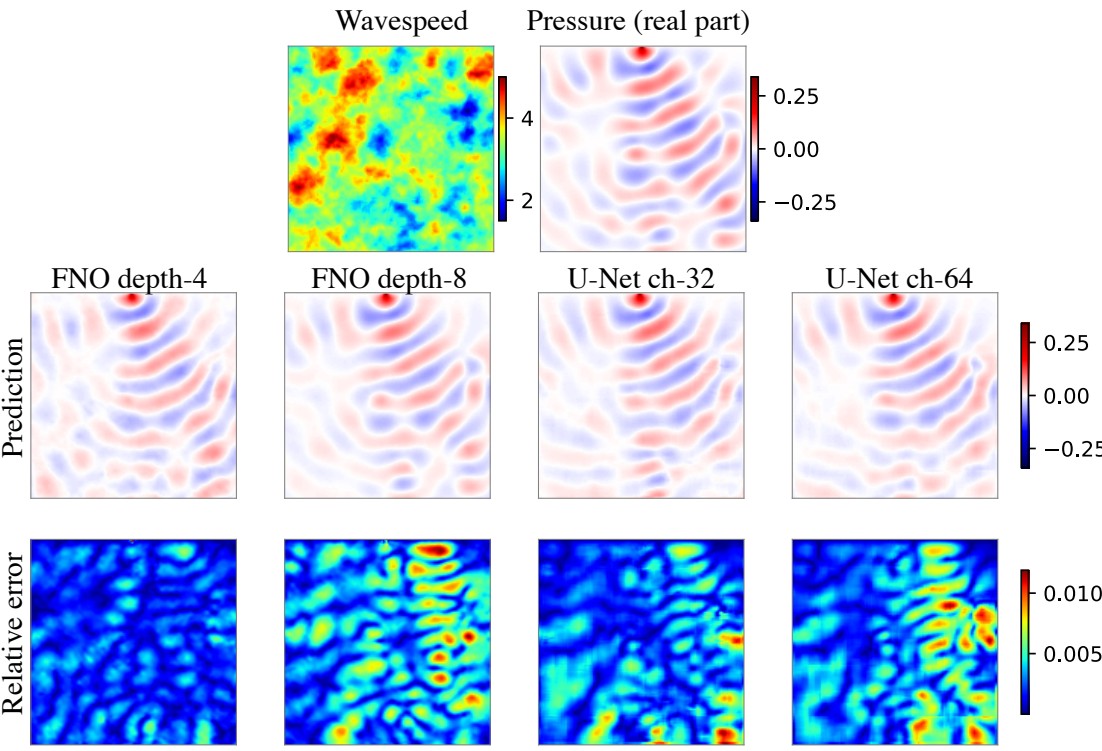

Figure 19: OOD test performance of models on a time-harmonic dataset. The model is trained on the time-harmonic dataset with *anisotropic* GRF wavespeed and frequency $\omega/2\pi = 15$Hz, but tested on the *isotropic* version instead. The figure layout follows from Figure 6 in the main text.

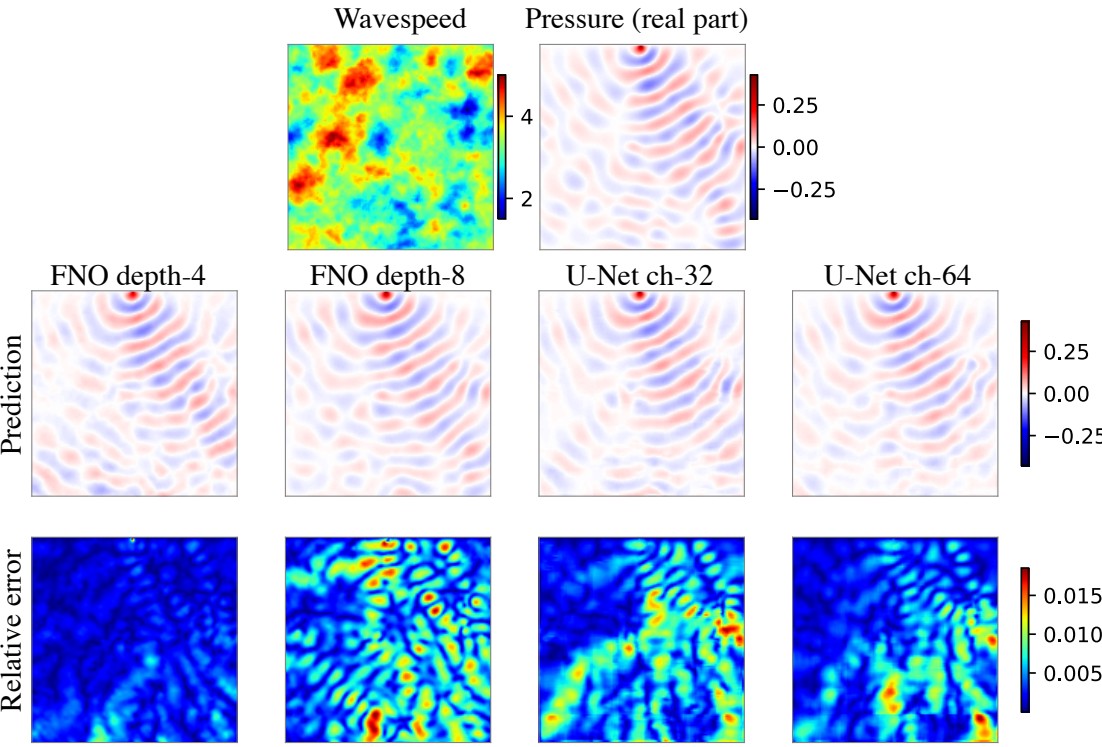

Figure 20: OOD test performance of models on a time-harmonic dataset. The model is trained on the time-harmonic dataset with *anisotropic* GRF wavespeed and frequency $\omega/2\pi = 20$Hz, but tested on the *isotropic* version instead. The figure layout follows from Figure 6 in the main text.

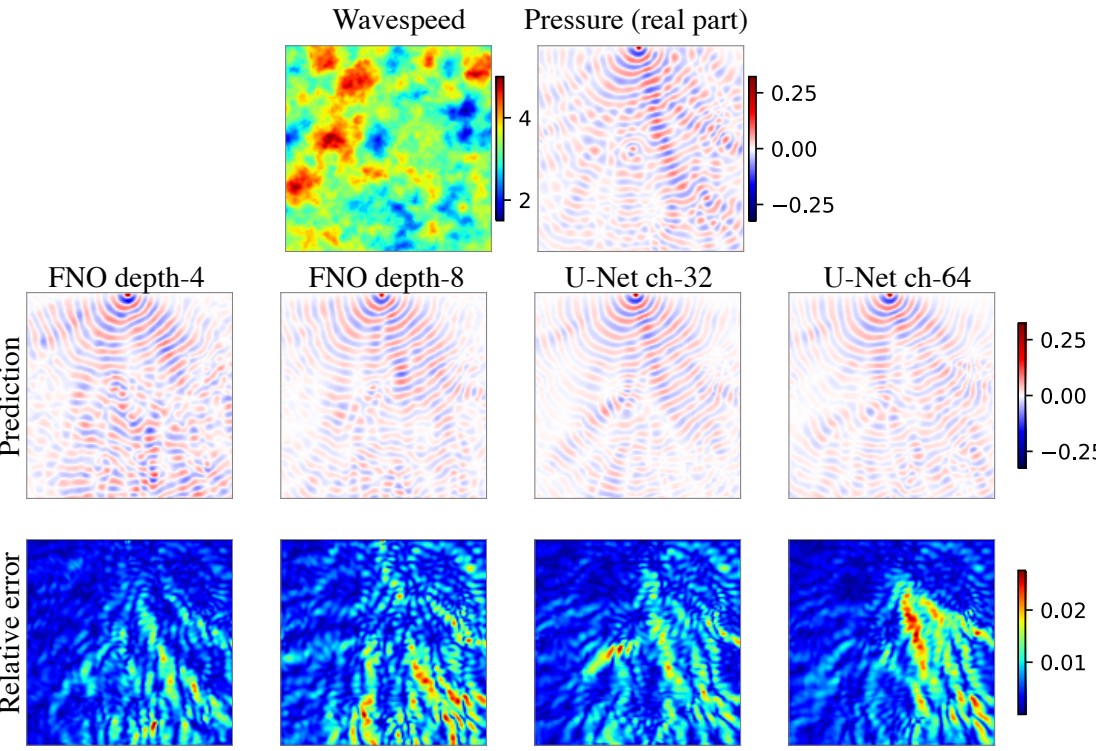

Figure 21: OOD test performance of models on a time-harmonic dataset. The model is trained on the time-harmonic dataset with *anisotropic* GRF wavespeed and frequency $\omega/2\pi = 40$Hz, but tested on the *isotropic* version instead. The figure layout follows from Figure 6 in the main text.

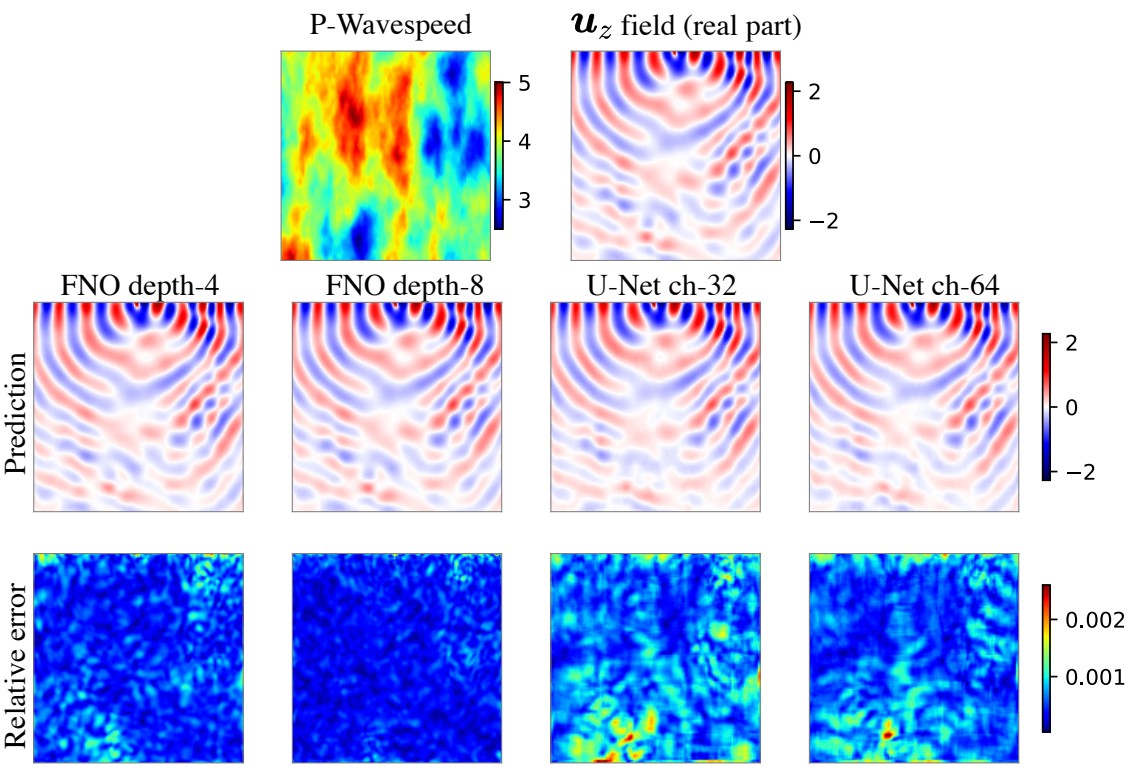

Figure 22: In-distribution test performance of models on an elastic time-harmonic dataset. The time-harmonic dataset is configured with anisotropic GRF wavespeed and frequency $\omega/2\pi = 10$Hz.

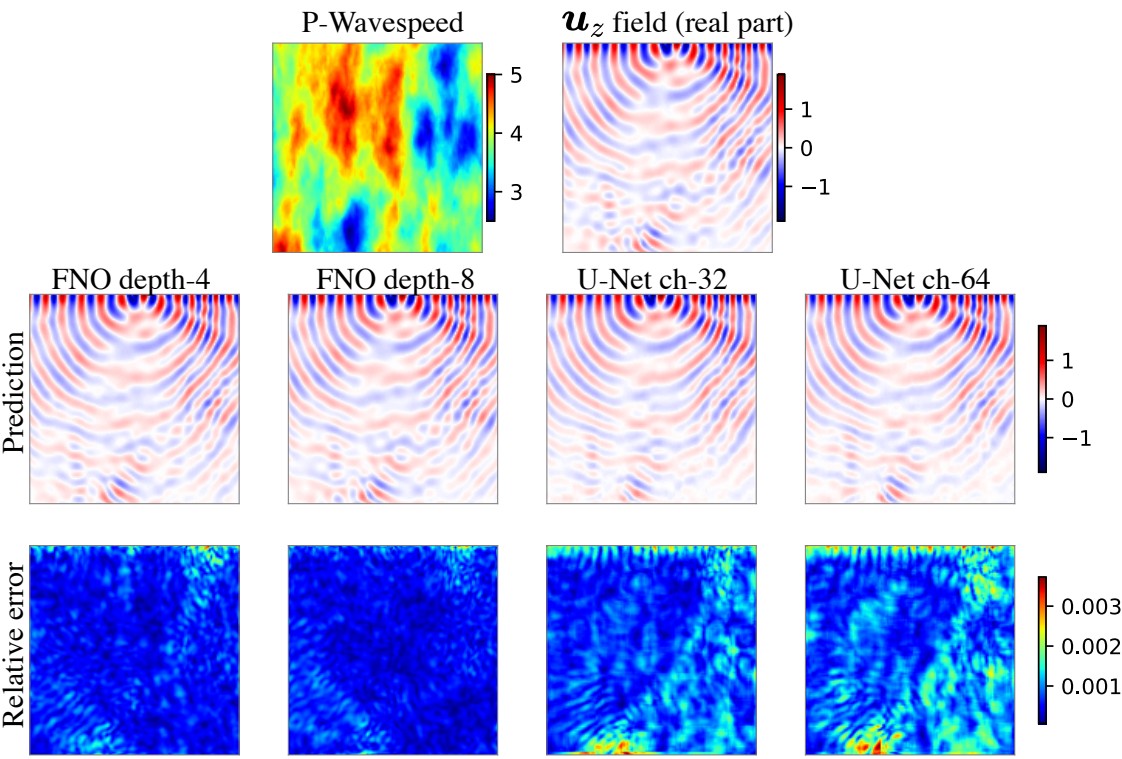

Figure 23: In-distribution test performance of models on an elastic time-harmonic dataset. The time-harmonic dataset is configured with anisotropic GRF wavespeed and frequency $\omega/2\pi = 15$Hz.

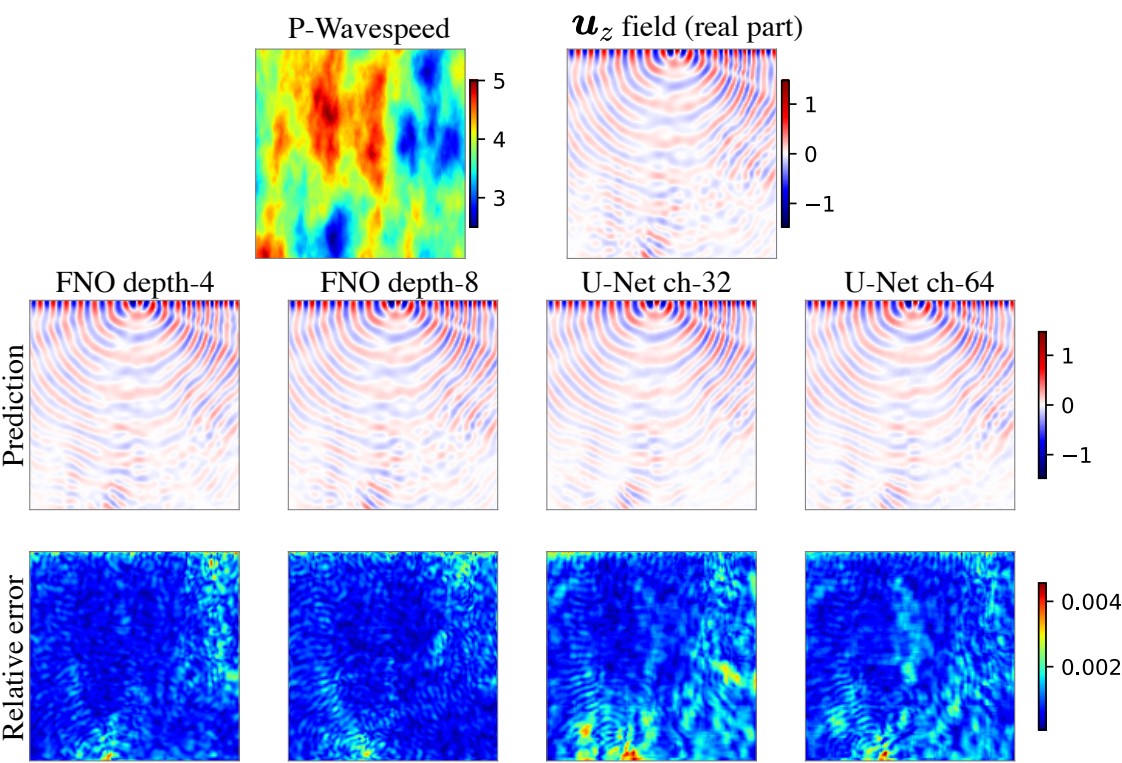

Figure 24: In-distribution test performance of models on an elastic time-harmonic dataset. The time-harmonic dataset is configured with anisotropic GRF wavespeed and frequency $\omega/2\pi = 20$Hz.

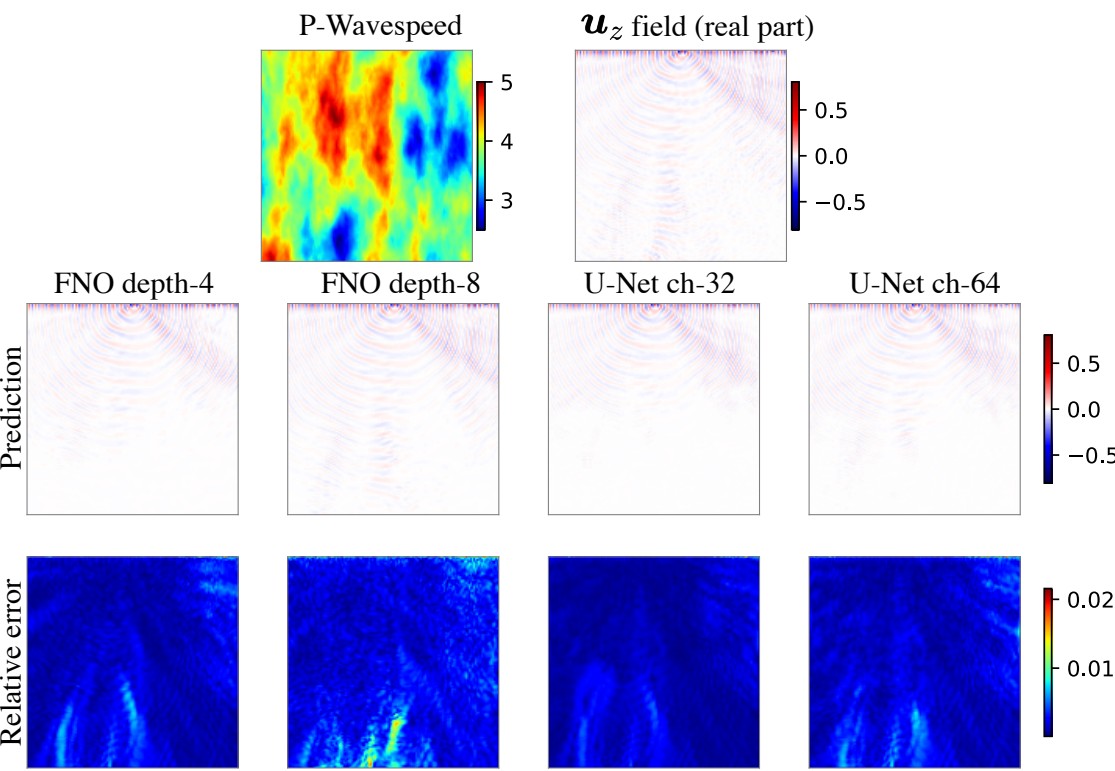

Figure 25: In-distribution test performance of models on an elastic time-harmonic dataset. The time-harmonic dataset is configured with anisotropic GRF wavespeed and frequency $\omega/2\pi = 40$Hz.

## D  Full experimental results of the time-varying datasets

| Problem | Wavespeed | Init. press. | FNO-depth-4 | FNO-depth-8 | U-Net-ch-32 | U-Net-ch-64 | UNO-modes-12 | UNO-modes-16 |
|---|---|---|---|---|---|---|---|---|
| RTC | Gaussian lens | Thick lines | 0.421 | 0.381 | 0.393 | 0.371 | 0.466 | 0.433 |
| | | MNIST | 0.461 | 0.410 | 0.525 | 0.520 | 0.478 | 0.434 |
| | Iso. GRF | Thick lines | 0.365 | 0.329 | 0.451 | 0.430 | 0.411 | 0.378 |
| | | MNIST | 0.342 | 0.349 | 0.535 | 0.517 | 0.337 | 0.329 |
| | Aniso. GRF | Thick lines | 0.348 | 0.308 | 0.432 | 0.414 | 0.395 | 0.362 |
| | | MNIST | 0.378 | 0.377 | 0.469 | 0.491 | 0.364 | 0.362 |
| IS | Gaussian lens | Thick lines | 0.550 | 0.500 | 0.446 | 0.443 | 0.551 | 0.545 |
| | | MNIST | 0.695 | 0.686 | 0.629 | 0.627 | 0.652 | 0.667 |
| | Iso. GRF | Thick lines | 0.436 | 0.380 | 0.383 | 0.356 | 0.462 | 0.441 |
| | | MNIST | 0.489 | 0.479 | 0.511 | 0.511 | 0.463 | 0.465 |
| | Aniso. GRF | Thick lines | 0.415 | 0.351 | 0.359 | 0.327 | 0.436 | 0.419 |
| | | MNIST | 0.471 | 0.449 | 0.537 | 0.514 | 0.413 | 0.414 |

Table 7: **Performance comparison of models on the test folds of the time-varying datasets.** The error metric is the relative L2 error $\|p - \widehat{p}\|_{L^2} / \|p\|_{L^2}$ between the ground-truth $p$ and prediction $\widehat{p}$.

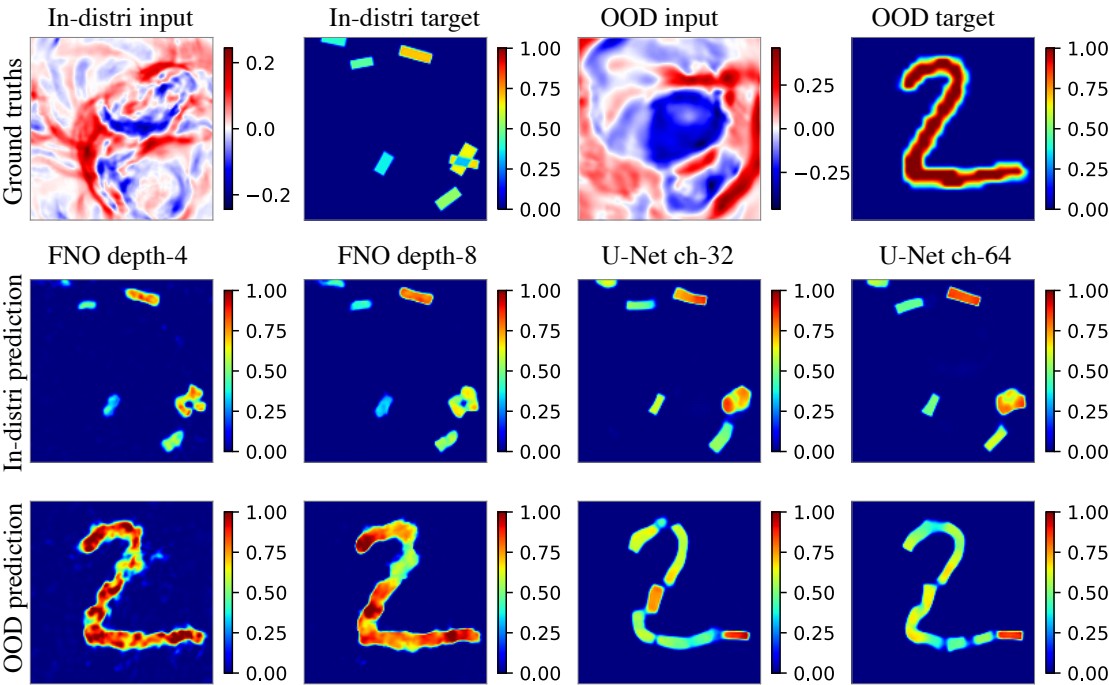

Figure 26: Test performance of models on the time-varying RTC dataset with isotropic GRF wavespeed. The first row shows the input and target samples of the RTC dataset; they can be either in-distribution or OOD. The second row shows the model predictions on the in-distribution sample. The second row shows the model predictions on the OOD sample.

# E    Case study: Gradually challenging the OOD generalization

We have seen the limitation of PDE surrogates on OOD samples. We note that the comparison "in-distribution vs OOD" is a simplified, binary notion, as it implies that a sample is either within a distribution or outside of it. More fine-grained OOD notions are helpful, as intuitively the performance of models on a sample may depend on the degree to which that sample differs from those in the training distribution.

In this section, we present a case study where we vary wavespeeds from near in-distribution samples ("less OOD") to distant ones ("more OOD"). See Figure 31 for visualization. We employ neural style transfer (NST) (Gatys et al., 2016) to create '0' digits of different OOD levels. These images are then used as wavespeeds in time-varying problems.

Recall that the NST algorithm separates and recombines the content and the style of images. Our content of interest is a '0' digit and the style of interest box-like strokes. The style and content of an image are balanced by a content weighting factor; see Gatys et al. (2016) for details. By adjusting the content weighting factor (referred to as OOD weight in Figure 31) from small to large, we generate images that resemble in-distribution thick lines (left panels of Figure 31(A)) and OOD MNIST (right panels of Figure 31(A)). We use the open-source NST implementation[1] for data generation.

Using different OOD degrees for wavespeed samples, we assess various PDE surrogates on RTC and IS tasks. Figure 31(B) displays the numerical results. As anticipated, higher OOD degrees' wavespeeds are harder to recover, giving higher relative errors.

---

[1]https://github.com/crowsonkb/style-transfer-pytorch

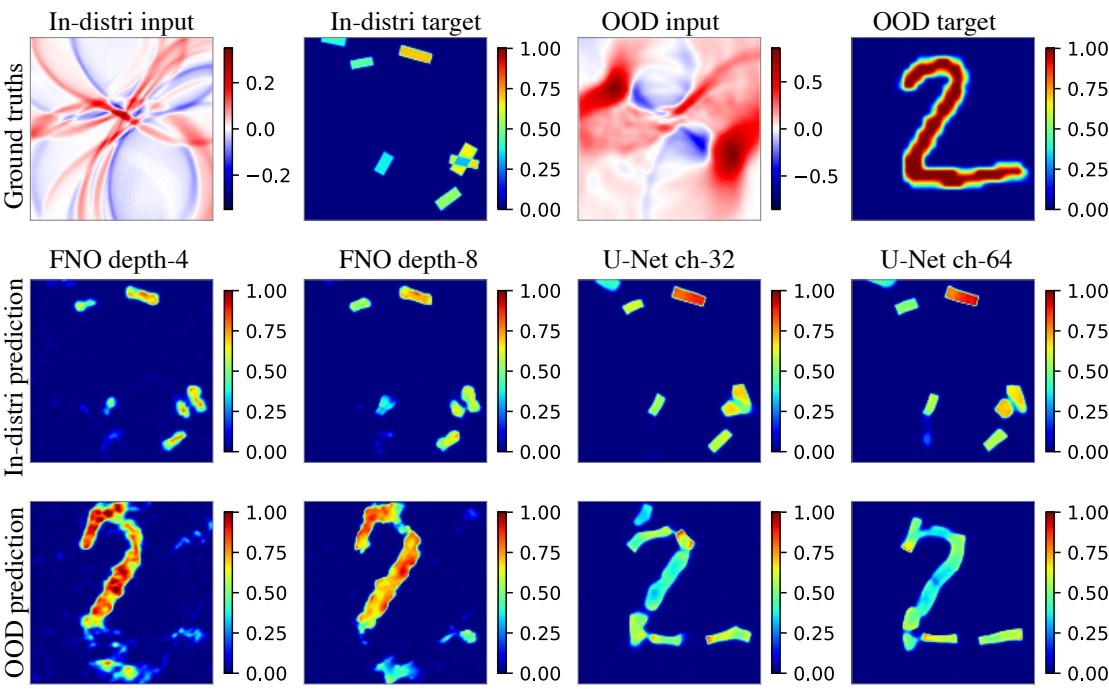

Figure 27: Test performance of models on the time-varying RTC dataset with Gaussian lens wavespeed. The figure layout is the same as Figure 26.

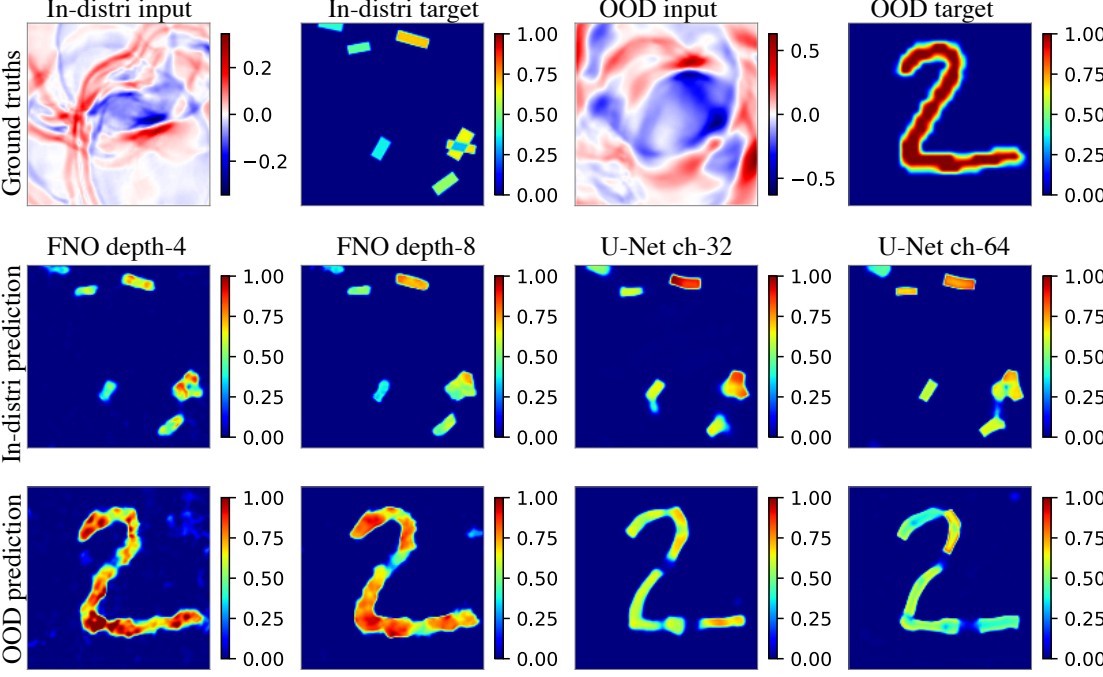

Figure 28: Test performance of models on the time-varying RTC dataset with anisotropic GRF wavespeed. The figure layout is the same as Figure 26.

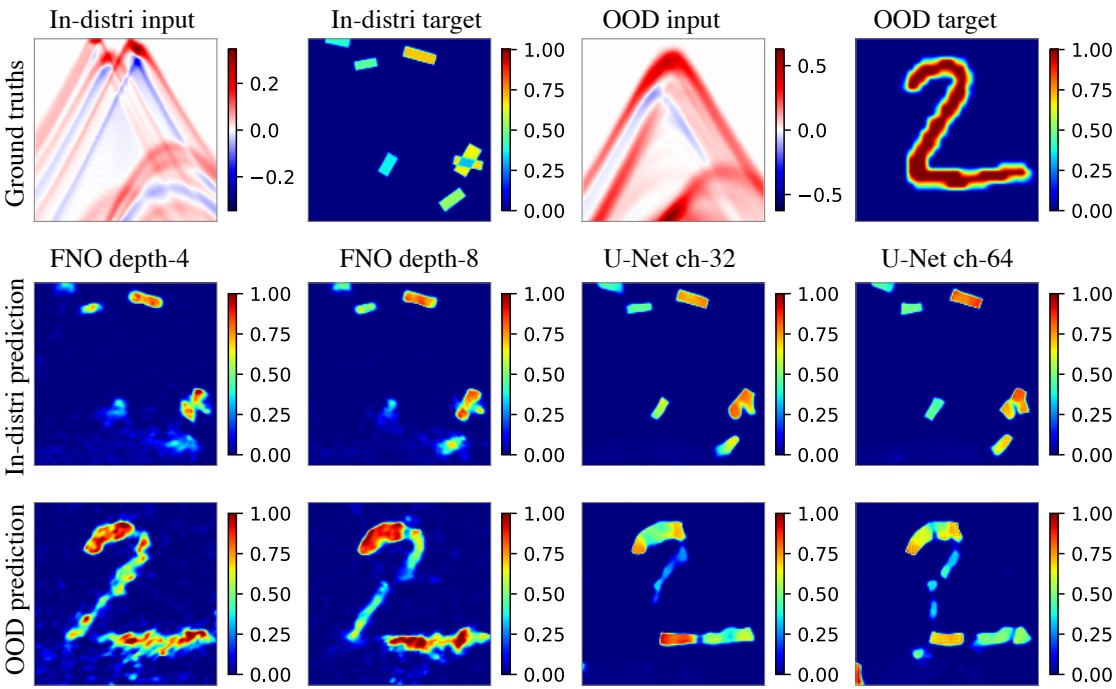

Figure 29: Test performance of models on the time-varying IS dataset with Gaussian lens wavespeed. The figure layout is the same as Figure 7.

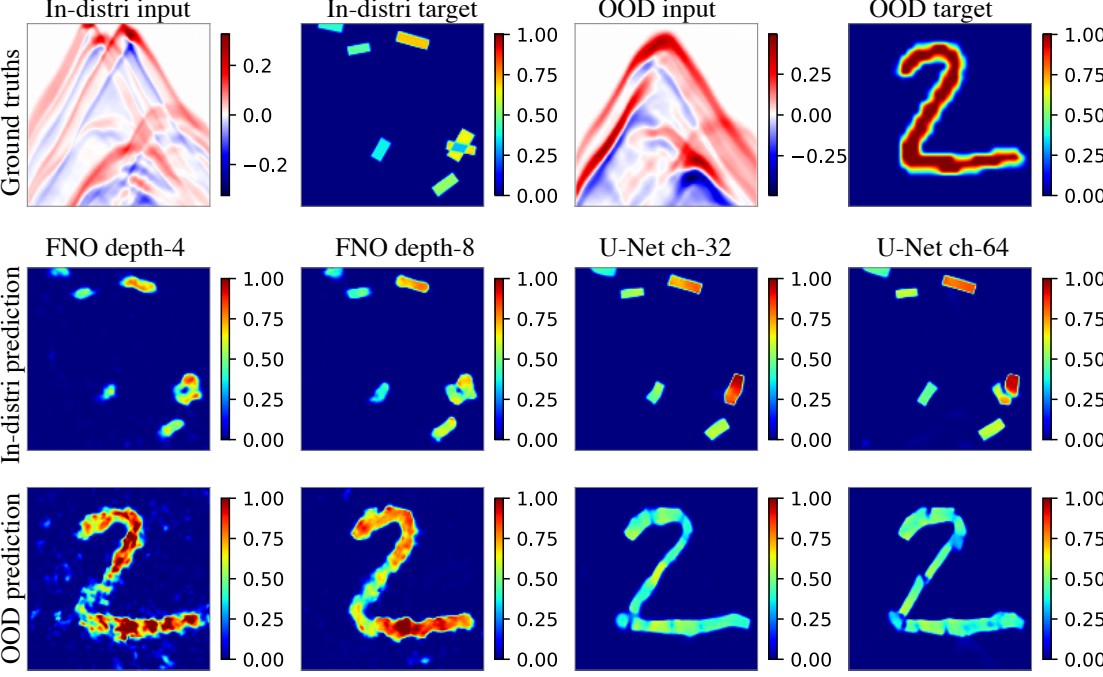

Figure 30: Test performance of models on the time-varying IS dataset with anisotropic GRF wavespeed. The figure layout is the same as Figure 7.

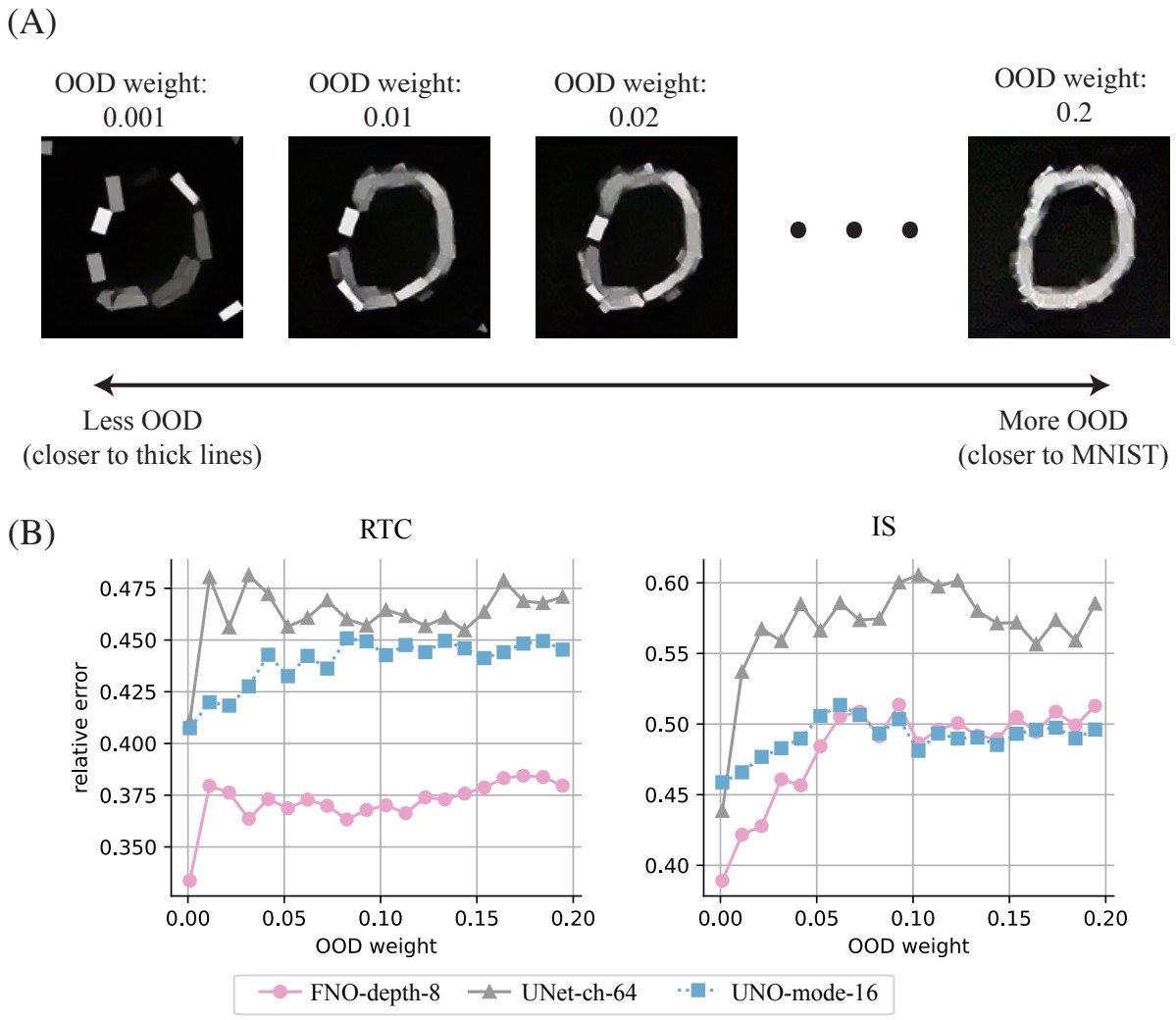

Figure 31: Case study: Model performance and OOD degree. Panel (A): Samples from neural style transfer, with content factor weights as 20 OOD weights ranging from 0.001 to 0.2. Panel (B): Models' performance on diverse wavespeed samples. More OOD wavespeed (closer to MNIST than thick lines) results in higher errors.

