# OpenReview forum: "WaveBench: Benchmarking Data-driven Solvers for Linear Wave Propagation PDEs"
_TMLR — Accepted by TMLR_

### Review · Reviewer_2kwj · 2023-10-19

**Summary Of Contributions:**

The paper introduces a new dataset for various forms of the wave equation. 	The author(s) provide a detailed description of how the data was generated, suggest example applications, and provide and discuss results for various state of the art models.

**Audience:**

Yes

**Broader Impact Concerns:**

No concerns.

**Claims And Evidence:**

Yes

**Requested Changes:**

- I could certainly live with a paper that is restricted in its applications. However, I would like more information perferably in the abstract (or if necessary in the introduction) about what kind of data is given and what kinds of problems are supposed to be solved by this data. See all points above, and this is me (not an expert about the wave equation) just thinking about spontaneously while writing this review. A lot more care should go into clearly describing the place of these datasets in the (wide!) field (in cs, engineering, math, and physics) about the wave equation. None of this literature is even hinted at, and virtually only ml papers are being cited.
 - Why do you introduce the Helmholtz equation before even writing down the wave equation? The connection between Helmholtz and wave equation is long known in math, but probably not so much in ML. Hence, a hint of their connection (or even the full derivation, which is not long) might be helpful.
 - title: When reading "wave propagation PDEs" I am not sure whether the linear wave equations are considered more the non-linear PDEs modeling e.g. ocean waves.
 - p. 2: "WaveBench includes 24 datasets" vs. page 3: "Our proposed WaveBench consists of 20 datasets"

**Strengths And Weaknesses:**

Stengths:
 - The paper is certainly interesting for a growing subcommunity in machine learning.
 - The machine learning part seems well done. Experiments are performed on reasonable baseline models and clearly described. The results are adequately discussed and strength and weaknesses of the baseline models are highlighted. These results are very believable.
 - The visualizations of the data are well done.
 - Details of the data generation are clearly.

Weaknesses:

There is one major weakness: The 24 datasets relating to the wave equation are far from covering all possible applications of the wave equation.
- The main restriction is to have Gaussian random fields as right hand sides. Some applications have zero right hand sides, which are not covered this way. Some applications have much stronger structured right hand sides.
- The restriction to d=2 might sound reasonable when trying to minimize computation time. However, ML methods for PDEs are supposed to shine in comparison to numerical solvers particularly in higher dimensions. Or the rather relevant Maxwell's equations need the wave equation in d=2.
- "Each component (ux or uy ) is complex-valued." This is a restriction, as the real wave equation is arguably at least as important as the complex one.
- For linear PDEs and the wave equation, there are some recent GP models, see later. These are asking different problems regarding the wave equations and it seems your datasets do not (obviously) cover their applications.
 1) Härkönen et al., Gaussian Process Priors for Systems of Linear Partial Differential Equations with Constant Coefficients, ICML 2023
 2) Henderson et al., Covariance models and Gaussian process regression for the wave equation. Application to related inverse problems, Journal of Computational Physics 2023
 3) Henderson et al., Wave equation-tailored Gaussian process regression with applications to related inverse problems, preprint
 4) Henderson et al., Stochastic Processes Under Linear Differential Constraints: Application to Gaussian Process Regression for the 3 Dimensional Free Space Wave Equation, preprint

There are some minor problems on the level of typos, see below.

---

> ### Author Response · Authors · 2023-11-08
> **Response from authors**
>
> Thank you for your review!
>
>
> > There is one major weakness: The 24 datasets relating to the wave equation are far from covering all possible applications of the wave equation. [...] I would like more information preferably in the abstract (or if necessary in the introduction) about what kind of data is given and what kinds of problems are supposed to be solved by this data
>
> We agree—waves are ubiquitous across scales, sciences, engineering applications, …, and different applications require different settings that cannot be covered by our 24 datasets. For instance, seismic imaging would require datasets with depth-varying layered wavespeeds, faulting, and inclusions such as salt bodies. For medical soft tissue ultrasound, datasets with small variations of wave speed would be more appropriate. Our goal with WaveBench is to target current identified deficiencies of learned wave solvers rather than cover all possible applications of the wave equation (or be hyperrealistic in a particular one). We demonstrate that, even with ample training data and relatively simple wave speed (Gaussian random field), PDE surrogates still face OOD generalization issues. But we agree that this point needs to be made more clear in our paper. Following your suggestion, we have edited the abstract to inform the readers about the PDE dimensionality and the kind of wave speed.
>
> > For linear PDEs and the wave equation, there are some recent GP models, see later. These are asking different problems regarding the wave equations and it seems your datasets do not (obviously) cover their applications.
>
> Thanks for bringing to our attention this line of work of GP models of PDE surrogates. Indeed, the questions asked in these papers are different, but they remain relevant to our machine learning for PDE solving theme. In our newly uploaded version, we have cited the two published papers for the interested readers.
>
> > The connection between Helmholtz and the wave equation is long known in math, but probably not so much in ML. Hence, a hint of their connection (or even the full derivation, which is not long) might be helpful.
>
> We have incorporated this suggestion in our newly updated version. In Section 3.1, we now provide references for detailed derivation.
>
>
> > title: When reading "wave propagation PDEs" I am not sure whether the linear wave equations are considered more the non-linear PDEs modeling e.g. ocean waves.
>
> Wave PDEs considered in our work are linear (although the solution operators for coefficien-to-solution maps or for inverse maps are not). We have added “linear” in the title of our updated paper to make this clearer.
>
> > p. 2: "WaveBench includes 24 datasets" vs. page 3: "Our proposed WaveBench consists of 20 datasets"
>
> Thanks for spotting it. It has been corrected in our latest version.

---

> ### Comment · Reviewer_2kwj · 2023-11-12
> **After reading the reviews**
>
> It seems to me, that all reviewers agree that the paper is generally interesting, but needs some work in the details. The points raised by the other reviewers are all good suggestions that should be taken serious by the author(s).
>
> For a recommendation regarding acceptance, I need to know form the authors how the points that were raised will be addressed.

---

> > ### Comment · Reviewer_2kwj · 2023-11-26
> > **Reply to my suggestions?**
> >
> > I have not gotten an answer to my review. It seems that some of my suggestions have been addressed in the recently uploaded pdf. However, I do not intent to go over the entire document again and check everything. Currently, I can only suggest weak reject, which would be a pity, since I guess a simple answer from the authors to my review might clarify everything.

---

> ### Author Response · Authors · 2023-11-26
> **Visibility mistake**
>
> Hi! We apologize for a mistake we made regarding the visibility of our previous post---we responded to your reviews a few weeks ago, but we set the post's visibility incorrectly. We have just corrected this mistake. Thank you for your prompt review and for bringing this to our attention! If you have questions and remarks regarding our response, we are more than willing to engage with you.

---

### Review · Reviewer_Bern · 2023-11-01

**Summary Of Contributions:**

A number of benchmarking datasets for surrogate learning for wave propagation PDEs are proposed. The collection comprises not only different wave PDEs but also multiple problem settings. Moreover, some baseline methods of operator learning are examined on these datasets.

**Audience:**

Yes

**Claims And Evidence:**

Yes

**Requested Changes:**

I would suggest no major changes. It would be great if the authors could check the minor points that I listed in the previous form and provide relevant information or make changes accordingly.

**Strengths And Weaknesses:**

The paper presents certainly a valuable contribution as a collection of datasets for PDE surrogate learning. The focus on the wave propagation PDEs sounds reasonable given the brief literature review in the paper (though I am not really qualified to assess how the current contribution is different from existing benchmarking datasets). The given datasets seem sufficiently challenging as operator learning problems.

Some minor points were unclear as listed below:

(1) For the time-harmonic elastic waves, what is the typical form of the operator $G^\dagger$ to be estimated? It is missing only for this particular case.

(2) Are there any reasons behind the particular sizes of the datasets provided? e.g., do they match with the numbers in other similar datasets?

(3) Is it correct that the codes for **generating** (and not benchmarking) the data will not be published? Or are they included in the "benchmark code" to be in a github repository?

---

> ### Author Response · Authors · 2023-11-08
> **Response from authors**
>
> Thank you for your review.
>
> > (1) For the time-harmonic elastic waves, what is the typical form of the operator
>  to be estimated? It is missing only for this particular case.
>
> Thank you for the comment, we have updated the manuscript to incorporate the former appendix A into the main text in order to clarify the equations and operators. The details for the elastic wave problem  are now all in the the subsection 3.1.2.
>
> > (2) Are there any reasons behind the particular sizes of the datasets provided? e.g., do they match with the numbers in other similar datasets?
>
> To our knowledge, our datasets are among the largest PDE datasets available. Typically, PDE datasets of the larger end contain around 10,000 samples, as seen in Table 1 of the PDEBench paper (Takamoto et al., 2022). We prepared 50,000 samples for each wave dataset to prevent large neural operator models from overfitting easily. However, if users want to work with fewer samples, such as for evaluating neural operators in data-limited scenarios, the dataset can be easily divided into smaller portions.
>
> > (3) Is it correct that the codes for generating (and not benchmarking) the data will not be published? Or are they included in the "benchmark code" to be in a github repository?
>
> Thank you for bringing this up! The code for generating the data is included. We will provide the GitHub link once the anonymous period concludes.

---

### Review · Reviewer_LgzV · 2023-11-11

**Summary Of Contributions:**

This submission presents a collection of dataset of wave propagation PDEs for time-harmonic wave equations and time-varying wave equations respectively.
The time-harmonic wave dataset concerns acoustic wave problems and elastic wave problems.
And the time-varying wave dataset concerns final-value problems where the termination condition is given as local or global information and the target is to infer the initial condition.

The source dataset is generated by hybridizable discontinuous Galerkin method implemented in ‘hawen’ package.

**Audience:**

Yes

**Claims And Evidence:**

Yes

**Requested Changes:**

Questions:


-- Why not submit to venues like NeurIPS 2023 Datasets and Benchmarks Track [https://nips.cc/Conferences/2023/CallForDatasetsBenchmarks]? This work will receive wider exposure to the ML community when published in a dataset-specific venue.

-- In equation (1), there is an external driving force term $f(\mathbf{x},\omega)$. There is no specification in the entire manuscript about this term.

-- Specify how the Gaussian random fields are generated for the wave speed $c(\mathbf{x})$.

-- What are $\lambda$ and $\mu$ in equations (8) and (9) in appendix section A.2?

-- Considering there are multiple hyperparameters in setting up the PDE problem, authors should make a table (for each subtype of problem, or for a general type) detailing the combination of the parameters.

-- Does the notation $\dagger$ imply adjoint operator in any case in the manuscript?

-- For section 4, what loss functions are used to set up training? There should be detailed explanation of training setup.

-- Can authors comment on if the solution accuracy results by the selected methods (FNO, Unet and UNO) are up to expectation considering reported performance in precedent work which also leverages these methods?

Minor comments:


Typo: section 2 Background, paragraph bold font: ‘bechmarks’

**Strengths And Weaknesses:**

Strengths:
-- Presentation of the main text is clear and straightforward.

Weaknesses:
-- Several technical details should be further clarified. Specific questions are stated in the “Requested Change” section.

---

> ### Author Response · Authors · 2023-11-19
> **Response from authors 1/2**
>
> Thank you for your review. Based on your suggestions, we made several improvements in our updated manuscript. Below are our responses point-to-point.
>
> > Why not submit to venues like NeurIPS 2023 Datasets and Benchmarks Track
>
> We did submit to NeurIPS 2023 Datasets and Benchmarks Track. Our scores were (7, 6, 6, 5), and we are happy to make the reviews and our rebuttal available. We significantly revised our work based on the reviewers' feedback during the rebuttal phase of NeurIPS 2023, but two reviewers didn't respond to our updates.
>
> We believe that wavebench is well-suited for TMLR.  TMLR has recently published number of papers related to learning forward and inverse surrogate operators:
>
> - [Towards Multi-Spatiotemporal-Scale Generalized PDE Modeling](https://openreview.net/pdf?id=dPSTDbGtBY)
> - [The U-shape neural operator](https://openreview.net/pdf?id=j3oQF9coJd)
> - [Generative Adversarial Neural Operators](https://openreview.net/pdf?id=X1VzbBU6xZ)
>
> and papers that put forward new datasets:
>
> - [WOODS: Benchmarks for Out-of-Distribution Generalization in Time Series](https://openreview.net/pdf?id=mvftzofTYQ)
> - [NovelCraft: A Dataset for Novelty Detection and Discovery in Open Worlds](https://openreview.net/pdf?id=4eL6z9ziw7).
> - [TimeSeAD: Benchmarking Deep Multivariate Time-Series Anomaly Detection](https://openreview.net/pdf?id=iMmsCI0JsS)
>
>
> > In equation (1), there is an external driving force term $f(x, \omega)$. There is no specification in the entire manuscript about this term.
>
> The source function f is a point-source, that is, a delta-Dirac in space $\delta(y)$ where $y$ is the position of the source. We have updated our manuscript accordingly to clarify this.
>
>
> > Specify how the Gaussian random fields are generated for the wave speed
>
> In our updated manuscript, we moved the description of the Gaussian random fields from the Appendix A1 to the main text for a clearer exposition. The generation procedure closely follows [(Benitez et al., 2023)](https://arxiv.org/pdf/2301.11509v2.pdf), where the random field is a composition of an affine transformation and a Gaussian random field with a covariance function. By setting the coefficient $\boldsymbol{\lambda} = (\lambda_x, \lambda_y)$ of the covariance, we obtain isotropic and anisotropic random fields.
>
> > $\lambda$ and $\mu$ in equations (8) and (9) in appendix section A.2?
>
> We have updated the manuscript to introduce the elastic equations and parameters directly in the main body (now in section 3.1.2). We have indicated that the parameters lambda and mu are the Lamé parameters that serve to define the P- and S-wavespeeds in equation 6.
>
> > Considering there are multiple hyperparameters in setting up the PDE problem, authors should make a table (for each subtype of problem, or for a general type) detailing the combination of the parameters.
>
> Thank you for this thoughtful suggestion. We have documented the varying hyperparameters across different datasets in Tables 2 and 3 in the Appendix. Initially, we have tried putting all hyperparameters, both fixed and varying across datasets, in a single table. But this made them a bit harder to interpret. For example, symbols such as $s$ and $\mathfrak{c}$ are not clear without additional context like the equations they appear in. To address your concerns, we have thoroughly revised the main text by drawing and refining content from the appendix. This revision allows readers to seamlessly understand experimental details without constantly referring back to the appendix, thereby enhancing the clarity of the experimental framework.
>
> > Does the notation $\dagger$ imply adjoint operator in any case in the manuscript?
>
> We use the notation $\dagger$ like in $G^\dagger$ to represent the ground-truth operator.
> That is, $G^\dagger$ is what our models try to learn from the data. The same notation was used in several prior works of PDE learning. See e.g., Equation 5 of [(Kovachki et al., 2022)](https://www.jmlr.org/papers/volume24/21-1524/21-1524.pdf), Equation (2.1) of [(de Hoop et al., 2022)](https://arxiv.org/pdf/2203.13181.pdf), and the text above Equation (1) in [(Li et al. 2021)](https://openreview.net/pdf?id=c8P9NQVtmnO).

---

> > ### Author Response · Authors · 2023-11-19
> > **Response from authors 2/2**
> >
> > > For section 4, what loss functions are used to set up training? There should be detailed explanation of training setup.
> >
> > The loss function used for training is the relative l2 loss, as reported in the “Training protocol” paragraph of Section 4.1: "We employed the relative L2 loss for training and evaluation in all our problems, following the approach in Li et al. (2021); de Hoop et al. (2022b)."
> > The same paragraph contains further training setup details. In case there is anything unclear in our description, we are happy to clarify. We reiterate that we will release all code, both for data generation and for model training.
> >
> > > Can authors comment on if the solution accuracy results by the selected methods (FNO, Unet and UNO) are up to expectation considering reported performance in precedent work which also leverages these methods?
> >
> > For in-distribution datasets, we anticipate that FNO, UNet, and UNO work well provided with sufficient amounts of training data. However, we are surprised by their limitations when it comes to OOD wave tasks even with sufficient amounts of training data—we believe that our work is the first that demonstrates the OOD limitation at scale.
> >
> >
> > >Minor comments:Typo: section 2 Background, paragraph bold font: ‘bechmarks’
> >
> > Thanks for spotting it. It is corrected in our newly uploaded submission.

---

### Decision · Action_Editor_Kaki · 2024-01-19

**Recommendation:** Accept as is

**Comment:**

The paper makes a serious effort to establish a first benchmark that will allow future assessments of algorithms for differential operator learning. As far as I know, a benchmark for the case of wave propagation does not exist despite its practical application.

**Audience:**

This benchmark will interest researchers working on machine learning for operator learning. A growing ML community may be interested in this benchmark, related code and algorithms.

**Claims And Evidence:**

The paper proposes a set of datasets related to wave propagation PDEs and tests several algorithms that attempt to learn the differential operator. Experiments are rigorously executed under different parameterisations for the PDEs. The claims are well supported by the experimental framework included in the paper.